**eLife** RESEARCH ARTICLE

# Deletion of the *moe*A gene in *Flavobacterium* IR1 drives structural color shift from green to blue and alters polysaccharide metabolism

Álvaro Escobar Doncel[1,2]*, Constantinos Patinios[3], Alexandre Campos[4], Maria Beatriz Walter Costa[2], Maria V Turkina[5], Maria Murace[6], Raymond HJ Staals[3], Silvia Vignolini[6,7], Bas E Dutilh[2,8], Colin J Ingham[7]*

[1]Hoekmine BV, Utrecht, Netherlands; [2]Institute of Biodiversity, Faculty of Biological Sciences, Cluster of Excellence Balance of the Microverse, Friedrich Schiller University Jena, Jena, Germany; [3]Laboratory of Microbiology, Wageningen University & Research, Wageningen, Netherlands; [4]CIIMAR, Interdisciplinary Centre of Marine and Environmental Research, Terminal de Cruzeiros do Porto de Leixões, Matosinhos, Portugal; [5]Department of Biomedical and Clinical Sciences, Faculty of Medicine and Health Sciences, Linköping University, Linköping, Sweden; [6]Yusuf Hamied Department of Chemistry, University of Cambridge, Cambridge, United Kingdom; [7]Max Planck Institute of Colloids and Interfaces, Am Mühlenberg, Germany; [8]Theoretical Biology and Bioinformatics, Science4Life, Utrecht University, Utrecht, Netherlands

**\*For correspondence:**
aescdon94@gmail.com (ÁED);
Colinutrecht@gmail.com (CJI)

**Competing interest:** The authors declare that no competing interests exist.

**eLife Assessment**

This manuscript presents **important** findings on how structural color can be manipulated through a specific single-gene mutation in a motile bacterium. **Compelling** data provide a promising model to identify genes and molecular mechanisms supporting this widespread optical phenomenon. This work will be of interest to biophysicists and microbiologists working on structural colors and Flavobacterium.

**Abstract** Structural colors (SC), generated by light interacting with nano-structured materials, are responsible for the brightest and most vivid coloration in nature. Despite being widespread within the tree of life, there is little knowledge of the genes involved. Partial exceptions are some *Flavobacteriia* in which genes involved in a number of pathways, including gliding motility and polysaccharide metabolism, have been linked to SC. A previous genomic analysis of SC and non-SC bacteria suggested that the pterin pathway is involved in the organization of bacteria to form SC. Here, we focus on *moe*A, a molybdopterin molybdenum transferase. When this gene was deleted from *Flavobacterium* IR1, the knock-out mutant showed a strong blue shift in SC of the colony compared to the wild-type. The *moe*A mutant showed a particularly strong blue shift when grown on kappa-carrageenan and was upregulated for starch degradation. To further analyze the molecular changes, proteomic analysis was performed, showing the upregulation of various polysaccharide utilization loci, which supported the link between *moe*A and polysaccharide metabolism in SC. Overall, we demonstrated that a targeted approach, modifying a single gene identified by genomics, could change the optical properties of bacteria.

**eLife digest** Nature never disappoints in its display of colourful organisms. A striking example is the iridescent plumage of the peacock, with its magnificent blue and green shading. But these fantastical colours are not produced by conventional dyes or pigments. As early as the 17th century, natural philosophers such as Robert Hooke and Isaac Newton discovered that they arise from light interacting with nanostructures in the feathers, which selectively reflect the intense blues and greens.

This optical phenomenon is known as structural colour, and it is widespread in nature – from flowers, seaweeds and seeds to many groups of animals. Structural colour serves a variety of functions, including light management, attraction of pollinators, photoprotection, and roles in sexual signalling, warning, or camouflage.

More recently, structural colours have also been observed in certain marine bacteria, particularly when they grow in dense groups or colonies. For example, the *Flavobacterium* strain IR1 isolated from Rotterdam harbour forms strikingly iridescent colonies when grown in the laboratory.

Despite these observations, little is known about why bacteria produce structural colour or how it is generated at the genetic level. To address this gap, Doncel et al. used a computational modelling approach to identify candidate genes potentially involved in structural colour formation in Flavobacterium IR1. Their analysis highlighted the *moe*A gene, which encodes the enzyme molybdopterin molebdenum transferase, as a likely candidate.

The researchers then experimentally deactivated *moe*A and assessed the effects using optics, proteomics and cultivation assays. Bacterial colonies lacking this gene showed a pronounced shift in colour from green to an intense blue. This change was traced to alterations in cell shape and morphology: in the absence of moeA, cells adopted a more elongated and regularly ordered shape, leading to a modified photonic structure (a change in the colony organisation) and a corresponding shift in reflected colour.

Further investigation of cellular processes affected by *moeA* revealed links to carbohydrate metabolism, particularly pathways associated with starch-like polysaccharides. Although the precise mechanistic role of *moeA* in regulating these processes remains to be fully elucidated, the results suggest that metabolic changes influence cell packing and, consequently, structural colour formation. Additional gene knockouts will be required to validate the proposed pathways and identify other genetic contributors.

Overall, this study demonstrates that genes play a direct role in shaping structural colour in bacteria. However, further research is needed to identify additional candidate genes in other structurally coloured species and to determine whether similar mechanisms operate across different organisms.

Beyond fundamental biology, structurally coloured bacteria also offer numerous possibilities. Differently coloured bacterial colonies could be harnessed as sustainable, bio-based materials. Indeed, colours made by cross-linking dead bacteria are already being used in art and design projects. In the future, such systems could potentially be scaled up and commercialised as environmentally friendly alternatives to synthetic pigments.

## Introduction

Structural color (SC) is the result of the interaction of light with nanoscale structures, causing selective, angle-dependent light reflectance, an optical mechanism distinct from pigmentation which is a property of differential light reflection in molecules. This phenomenon can have a bright, metallic, and iridescent appearance, where the color seen is often highly dependent on viewing and illumination angles. SC has been reported in many eukaryotes, including vertebrates, invertebrates, plants, and *Myxomycota*, as well as in bacteria, but not in *Eumycota* or *Archaea* (*Brodie et al., 2021*). Among bacteria, SC from colonies of the phylum Bacteroidetes is the best characterized (*Kientz et al., 2012b*; *Kientz et al., 2016*; *Johansen et al., 2018*). SC in bacteria results from the periodic organization of the rod-shaped cells packed in a regular hexagonal lattice, forming a two-dimensional photonic crystal that reflects light (*Schertel et al., 2020*). The ecological role of bacterial SC is yet to be determined. Hypotheses point at predation (*Hamidjaja et al., 2020*) and polysaccharide metabolism optimization (*van de Kerkhof et al., 2022*), but further research is needed to elucidate its biological significance.

Information on genes and pathways involved in bacterial SC is limited but growing. Transposon mutagenesis suggests the involvement of cellular functions including the stringent response, plant metabolite modification, carbohydrate metabolism, and Bacteroidetes-specific gliding motility (*Johansen et al., 2018*). A recent bioinformatic study has shown the possible link of some metabolic pathways, such as carbohydrate, pterin, and acetolactate metabolism, to bacterial SC (*Zomer et al., 2024*). In *Flavobacterium* iridescence species 1 (IR1), SC has been linked to interactions with microalgae, particularly through the metabolism of algal polysaccharides such as kappa-carrageenan and fucoidan (*Johansen et al., 2018*; *van de Kerkhof et al., 2022*). IR1's colony organization, which underlies SC, may play a role in interbacterial competition, such as predation, but this has no obvious link to the photonic properties of the bacteria (*Hamidjaja et al., 2020*).

A large-scale, genomic-based analysis of 117 bacteria strains (87 with SC and 30 without) identified genes potentially involved in SC by comparing gene presence/absence, providing an SC score (*Zomer et al., 2024*). By this method, pterin pathway genes were strongly predicted to be involved in SC. Pterins mainly work as enzyme cofactors in various functions, such as aerobic/anaerobic metabolism and detoxification. In eukaryotes, pterins contribute to pigmentary colors, such as in the scale structures of pierid butterfly wings (*Wijnen et al., 2007*), and appear in insects, fish, amphibians, and reptiles (*Colette Daubner and Lanzas, 2018*). While pigment coloration is different from SC, structurally organized pterins can function as refractive index dopants (*Wilts et al., 2017*; *Sai et al., 2023*) and function in UV protection, phototaxis, and intracellular signaling (*Feirer and Fuqua, 2017*).

We focused on one specific pterin, the molybdenum cofactor (MoCo), due to its predicted involvement in bacterial SC (*Zomer et al., 2024*). MoCo is a cofactor in a group of enzymes known as molybdoenzymes which are key enzymes in nitrogen, purine, and sulfur metabolism. These enzymes in bacteria fall into three families: xanthine oxidases, dimethyl sulfoxide reductases, and sulphite oxidases

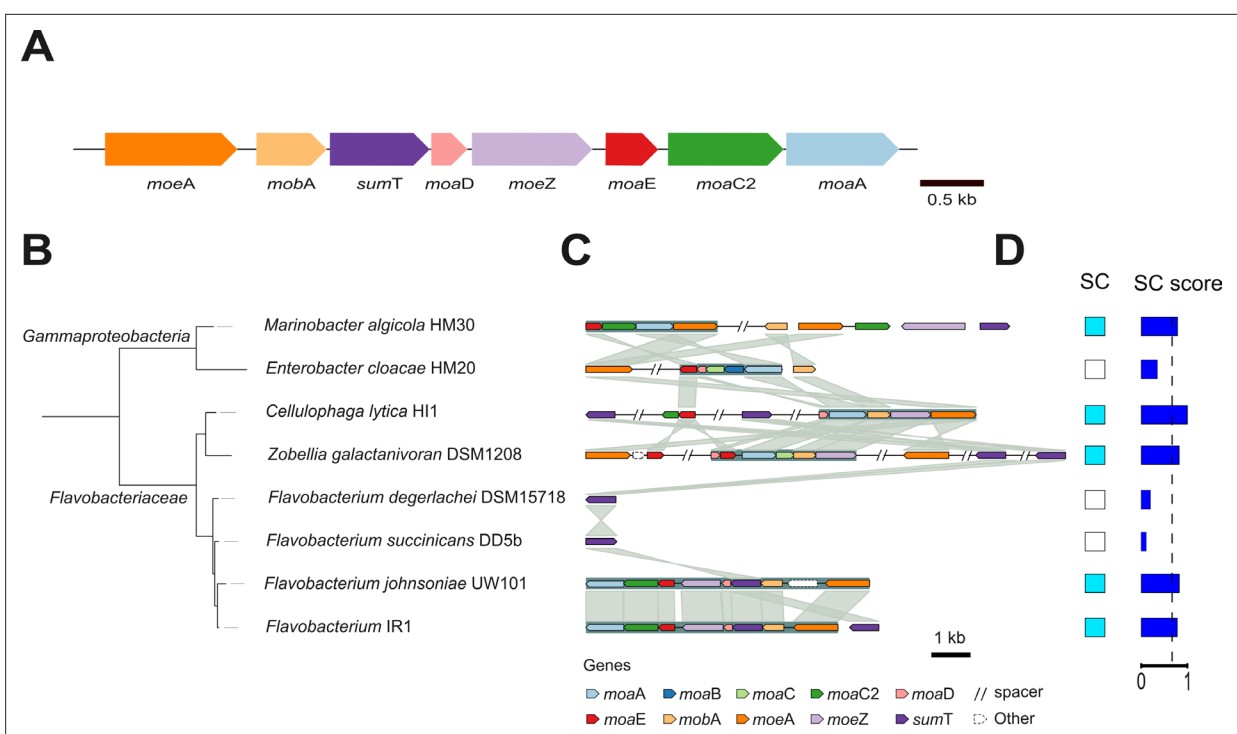

**Figure 1.** Phylogenetic analysis of MoCo operon. (**A**) Schematics of the putative molybdopterin synthesis operon in the IR1 genome. In blue, the target gene: *moe*A. (**B**) Phylogenetic tree of the 16 S ribosomal RNA gene, showing IR1 and other seven selected strains. (**C**) Synteny and homology visualization of genes that are putatively involved in molybdopterin synthesis. Spacers indicated with // represent stretches longer than 5 kb on the same contig, which may encode unshown genes. Whitespaces separate different contigs. (**D**) Presence of SC in the selected strains and its SC score based on the SC classifier software (*Zomer et al., 2024*). The suggested cut-off value (0.68) for presence of SC is shown as a dashed vertical line.

The online version of this article includes the following source data for figure 1:

**Source data 1.** Synteny coordinates of MoCo operon.

**Source data 2.** Table containing presence or absence of the MoCo synthesis genes in different bacterial genomes.

**Table 1.** Analysis of 117 bacterial genomes (87 SC and 30 non-SC) for the presence of the genes involved in molybdopterin cofactor synthesis.

| | *moe*A | *mob*A | *sum*T | *moa*D | *moe*Z | *moa*E | *moa*C2 | *moa*A |
|---|---|---|---|---|---|---|---|---|
| SC bacteria | 100% | 87% | 100% | 70% | 100% | 100% | 100% | 100% |
| Non-SC bacteria | 40% | 37% | 63% | 20% | 70% | 40% | 40% | 50% |

(*Wootton et al., 1991*; *Zhang and Gladyshev, 2008*). To study the link between MoCo and SC, we use IR1 as a model organism for bacterial SC due to the availability of genome engineering tools and its intense coloration (*Johansen et al., 2018*; *Patinios et al., 2021*). Using the SIBR-Cas (Self-splicing Intron-Based Riboswitch-Cas) genome engineering tool (*Patinios et al., 2021*), we deleted the molybdopterin molybdenum transferase *moe*A gene, one of the most important genes for predicting bacterial SC (*Zomer et al., 2024*), as its protein is crucial in the final MoCo pathway reaction.

## Results

### Bioinformatic analysis of the molybdopterin operon

We reanalyzed a recent bioinformatic analysis on SC to specifically investigate genes involved in molybdopterin cofactor (MoCo) synthesis (*Zomer et al., 2024*). Genes for MoCo synthesis are typically clustered in SC bacteria and are consecutively encoded on the IR1 genome (*Figure 1A*), probably forming an operon containing molybdopterin molybdenum transferase (*moe*A), molybdenum cofactor guanylyl transferase (*mob*A), uroporphyrinogen-III C-methyltransferase (*sum*T), molybdopterin synthase sulfur carrier unit (*moa*D), adenylyl transferase/sulfur transferase (*moe*Z), molybdopterin synthetase catalytic unit (*moa*E), cyclic pyranopterin monophosphate synthase 2 (*moa*C2), and GTP 3′,8-cyclase (*moa*A). In 117 bacterial genomes (87 SC and 30 non-SC) analyzed (*Table 1*), most bacteria showing SC contained all these genes, except *mob*A and *moa*D. Meanwhile, in non-SC bacteria, these genes appeared less frequently. Overall, 61 of 87 SC genomes had a complete MoCo pathway, 10 lacked one gene, and 16 lacked two. Conversely, only 6 of 30 non-SC genomes had a full pathway, while others showed partial gene loss, with 6 missing the entire pathway.

The genetic structure of this putative operon for molybdopterin synthesis was compared across eight strains with variable SC (*Figure 1BC*). Using the SC classifier, these strains were scored for SC based on the presence/absence of specific genes in their genomes (*Zomer et al., 2024*), revealing that the predictions were consistent with our experimental results SC (*Figure 1D*).

Synteny analysis of selected genomes revealed the organization of MoCo synthesis genes. IR1 contains a putative MoCo synthesis operon consisting of *moe*A, *mob*A, *sum*T, *moa*D, *moe*Z, *moa*E, *moa*C2, and *moa*A, and an additional *sum*T homolog. UW101 has a similar operon without the *sum*T duplication (*Figure 1C*). Other strains show different gene orders, loci, or variations like missing or duplicated genes. Notably, *Flavobacteriaceae* strains DSM15718 and DD5b, which only contain *sum*T, and HM20, lacking *sum*T but retaining most MoCo genes, do not exhibit SC. Thus, while the MoCo synthesis pathway is crucial for SC, its structure and organization vary among SC strains and are not the sole determinants of SC.

### Phenotyping the Δ*moe*A mutant

To study the role of *moe*A in SC, we generated a clean knock-out (KO) of *moe*A in IR1 using the SIBR-Cas tool (*Patinios et al., 2021*). After successfully deleting *moe*A, we compared the colors of the Δ*moe*A colonies with those of the wild-type (WT) strain under three nutrient conditions: (1) ASWB, a standard peptone/yeast extract medium; (2) ASWBLow, a low-nutrient medium with yeast extract as the sole nutrient; and (3) minimal medium (MM), with the minimum nutrients required for IR1 growth.

On ASWB agar plates, the WT strain colony showed a vivid brilliant green SC with a red ring, while the Δ*moe*A colony displayed a dull green-blue SC with a blue ring. On ASWBLow, the WT's SC shifted to a shiny green-yellow-orange color, whereas Δ*moe*A displayed a dull green center with an intense green ring. On MM, both the WT and Δ*moe*A had weaker SC than when grown on higher nutrient media, showed dispersed clusters of cells, and maintained their green and blue hues, respectively. Additionally, Δ*moe*A colonies spread more slowly than the WT under all conditions evaluated. In

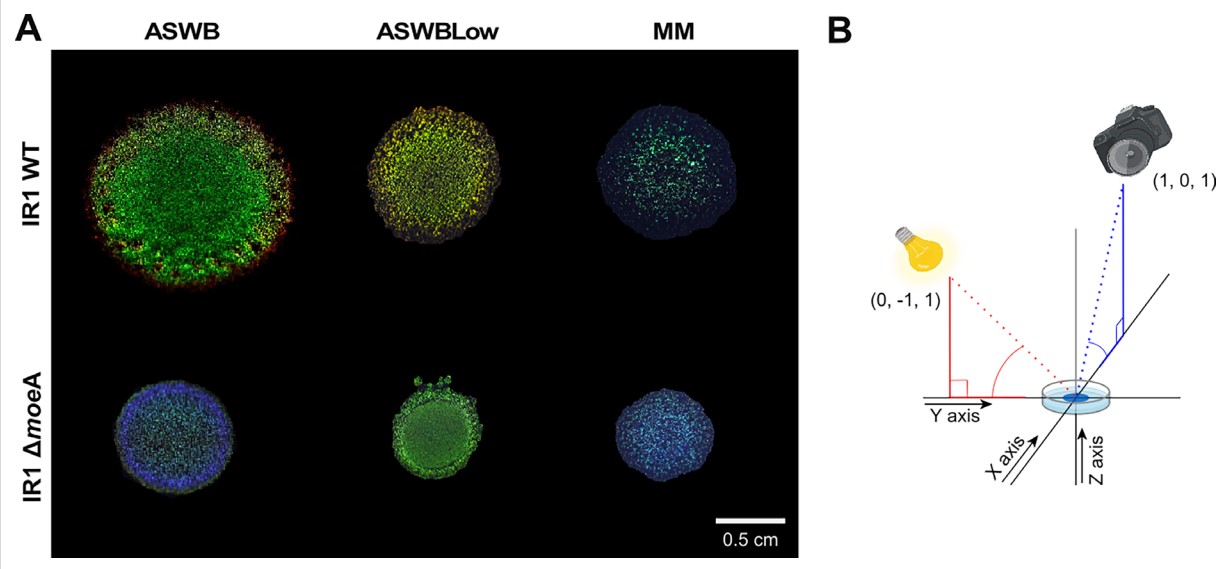

**Figure 2.** Optical phenotype of IR1 WT and ΔmoeA. (**A**) Colonies of IR1 WT and IR1 ΔmoeA grown on agar plates with three different nutrient conditions: ASWB, ASWBLow, and MM. (**B**) Schematic of how the colony image was taken. It shows the position of the incident light and the camera as X, Y, and Z coordinates (X,Y, Z). The colony is positioned at position (0,0,0), the light source at (0,−1,1), and the camera at (1,0,1). The red dotted line represents the light direction, the blue dotted line represents the camera direction, and the red and blue lines represent the position of the light and the camera.

summary, deleting *moe*A produced a general SC shift from green to blue and a reduction of colony spreading (*Figure 2A*).

The optical properties of the Δ*moe*A colony were checked by growing as a spot on ASWB plates and observing its color from different angles to capture the full optical response of its photonic structure. When photographed from directly above the light source at position (X,Y,Z coordinates 0,−1,1.1, respectively), Δ*moe*A displayed a primarily green SC (*Figure 3A*), albeit duller than when photographed from positions (1,-1,1; *Figure 3B*), and (1,−1,0.36 *Figure 3C*). From the positions in *Figure 3BC*, two distinct colored rings were visible: an inner blue ring and an outer green-yellow ring. Although when they were photographed from positions (1,0,0.36; *Figure 3D*), and (1,0,0.18; *Figure 3E*), the SC shifted to predominantly blue, with a highly reflective blue ring and a green ring. SC was lost when photographed from position (0.58,1,1), displaying a gray-brown color (*Figure 3F*). Overall, we confirmed the angle-dependency of the SC in the Δ*moe*A, showing variations in color and intensity with changes in viewing angle.

When IR1 was grown with the polysaccharides fucoidan (from brown algae), or kappa-carrageenan (from red algae), its SC shifted to dark purple and shinier green, respectively (*van de Kerkhof et al., 2022*). To investigate how nutrient supply affects SC in IR1 WT and Δ*moe*A strains, they were grown as spots on ASW medium gelled with kappa-carrageenan instead of agar (ASWBKC), fucoidan and agar (ASWBF), or starch and agar (ASWBS). On ASWBKC plates, both strains exhibited more intense SC than on ASWB, with Δ*moe*A displaying a brilliant, blue-shifted color compared to the WT's structural green. The WT strain also displayed a dark green ring and a thin red outer ring as observed in ASWB (*Johansen et al., 2018*; *Hamidjaja et al., 2020*). On ASWBF plates, the WT displayed a dull blue-purple SC, while Δ*moe*A showed a dull green SC with a dull green-yellow ring and a red thin outer ring. On ASWBS, the colonies displayed a mix of colors rather than the mostly monochromatic patterns seen on agar (*Figure 2*), kappa-carrageenan, or fucoidan (*Figure 4A*). The WT showed a dull green center, a green-yellow ring, and a shiny red outer ring. In contrast, Δ*moe*A displayed a dull blue center with a shiny blue ring and a shiny green outer ring. Overall, polysaccharides significantly influenced SC, with both strains showing the most intense colors on kappa-carrageenan.

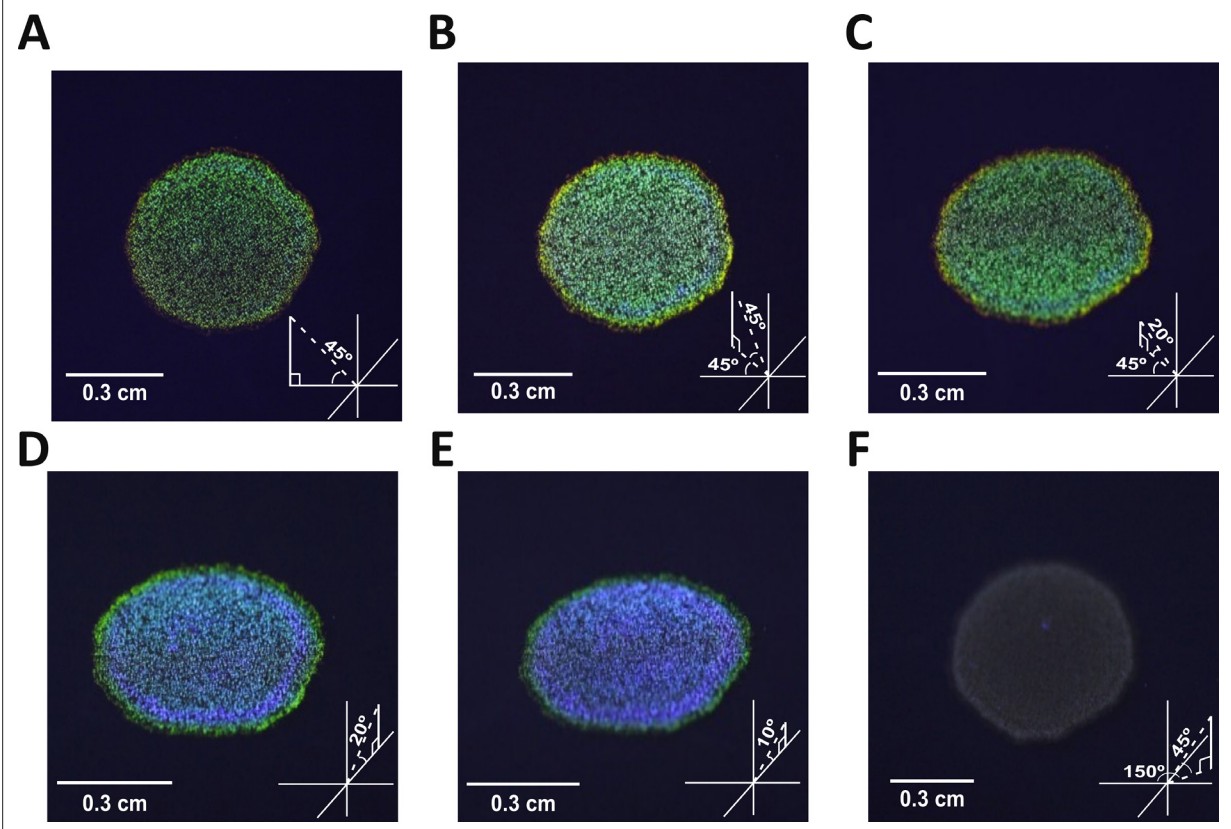

**Figure 3.** Δ*moe*A colonies grown on ASWB and photographed from different angles. The location of the camera is shown in the bottom right of each panel, following the scheme on **Figure 2B**. The camera coordinates are (**A**) (0,–1,1.1), (**B**) (1,-1,1), (**C**) (1,–1,0.36), (**D**) (1,0,0.36), (**E**) (1,0,0.18), and (**F**) (0.58,1,1). The light was always positioned at (0,–1,1).

## Quantification of the optical responses of IR1 WT and Δ*moe*A colonies

IR1 WT and Δ*moe*A grown on ASWBKC plates were studied using an optical goniometer to understand the optical characteristics of their displayed colors. We selected this media due to the uniform, vibrant blue coloration of the Δ*moe*A colony.

The complex optical response of both IR1 strains observed in the heatmaps in **Figure 5** can be attributed to a polycrystalline two-dimensional structure with hexagonal packing, as previously described (**Schertel et al., 2020**). In particular, the specular reflection data (**Figure 5A–B**) allowed us to extrapolate an effective refractive index of 1.38 for both strains, consistent with earlier studies (**Schertel et al., 2020**). In a diffraction configuration, intense diffraction peaks are observed in the visible range around a detection angle of –30° for wavelengths of 550 nm (green) for IR1 WT colonies (**Figure 5C**) and 480 nm (blue) for Δ*moe*A (**Figure 5D**), coherent with the primary colors observed qualitatively in **Figure 2A**.

In addition, two other bright diffraction spots are present in both cases outside of the visible range. For IR1 WT, such spots are present around 550 nm, 400 nm, and 350 nm; in Δ*moe*A, these diffraction spots shift to a lower wavelength around 480 nm, 350 nm, and 300 nm. By matching the diffraction grating equation with the observed spots (white dashed lines in **Figure 5**), the inter-bacterial distance can be obtained (**Schertel et al., 2020**). The periodicity was therefore estimated to be 410 nm for IR1 WT and 365 nm for Δ*moe*A. This optical analysis aligns with visual observations, confirming the blue shift in Δ*moe*A and suggesting that this change in SC is caused by cells which are likely to be narrower based on the estimated periodicity from the optical analysis.

## Deletion of the *moe*A gene reduces colony expansion

During the analysis of the colors displayed by IR1 WT and Δ*moe*A, differences in the colony spreading were observed indicating variations in gliding motility. To quantify this, both strains were grown for

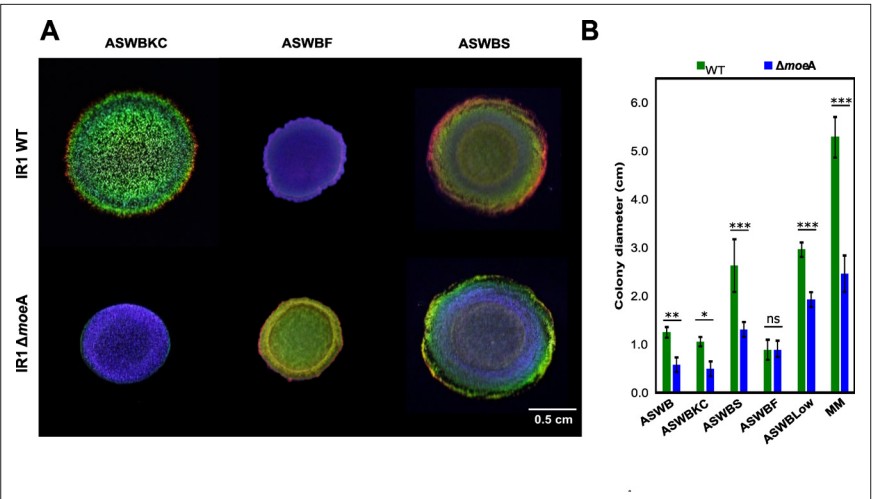

**Figure 4.** Influence of different polysaccharides in the colony color and diameter of IR1 WT and ΔmoeA. (**A**) Colonies of IR1 WT and ΔmoeA are grown for 2 days with 1% of three different polysaccharides: Artificial Sea Water Black with Kappa-Carrageenan instead of agar (ASWBKC), ASWB with agar and Fucoidan (ASWBF), and ASWB with agar Starch (ASWBS). All the photos were taken from position (1,0,1), following the scheme on **Figure 2B**. (**B**) Colony diameter in centimeters of IR1 WT and ΔmoeA grown on different media after 6 days, as mean ± standard deviation of three biological replicates. ns (non significance), * (p-value < 0.05), ** (p-value < 0.001), *** (p-value < 0.0001).

The online version of this article includes the following source data for figure 4:

**Source data 1.** Colony diameter data of IR1 WT and ΔmoeA when growing on different polysaccharides.

an extended period, and colony expansion was measured (**Figure 4B**). The ΔmoeA showed slower colony expansion, reaching about half the size of the WT in most conditions, except on ASWBF, where colony expansion was similar to the WT. Interestingly, ΔmoeA colony expansion was faster on ASWBLow, and especially on MM, compared to other conditions, which also happened for the WT. Thus, the lack of nutrients is an enhancer of colony expansion.

The organization and motility of groups of cells at the colony edges were visualized using a digital stereo microscope with full coaxial light. Both strains were grown as a spot on ASWB, and the colony edges were visualized for 1 hr (**Figure 6**). IR1 WT showed high motility of the bacterial layers at the edge of the colony, with dispersed cell layers forming 'vortex' patterns (**Figure 6**, yellow arrows). In contrast, ΔmoeA exhibited limited motility, with a more tightly packed cell organization and a fine, slow-moving layer at the edge (**Figure 6**, blue arrows), and did not show a 'vortex' pattern. This suggests that moeA deletion significantly impairs cell motility and colony expansion.

## Changes in the proteome due to the deletion of *moe*A

To further investigate the effects of moeA deletion, we performed a characterization and quantitative comparison of cellular (**Figure 7A**) and extracellular (**Figure 7B**) proteomes of IR1 WT and ΔmoeA strain using a mass spectrometry-based proteomic approach. We identified 203 intracellular proteins that significantly changed their abundance upon deletion of moeA (**Supplementary file 1d and e**), and 268 differentially abundant extracellular proteins (**Supplementary file 1f and g**). The following pathway analysis provided insight into how these proteins might be related to SC.

Peptides derived from molybdopterin molybdenum transferase, encoded by moeA, were only detected in the WT strain, confirming a successful knockout in ΔmoeA. The intra- and extracellular proteome analysis showed some differentially expressed proteins involved in the MoCo pathway or containing molybdopterin-binding motif. The deletion of moeA produced different regulatory effects on the peptides encoded from the genes within its putative operon. The proteins encoded from moaA, moaC2, and mobA were upregulated in the mutant, while those from moaE and moeZ were unaffected, and those from sumT and moaD were undetected in both strains (**Figure 8**). Proteins with a molybdopterin-binding motif were differentially expressed. The downregulated proteins included xanthine dehydrogenase yagS and yagR (involved in purine catabolism), an alanine dehydrogenase

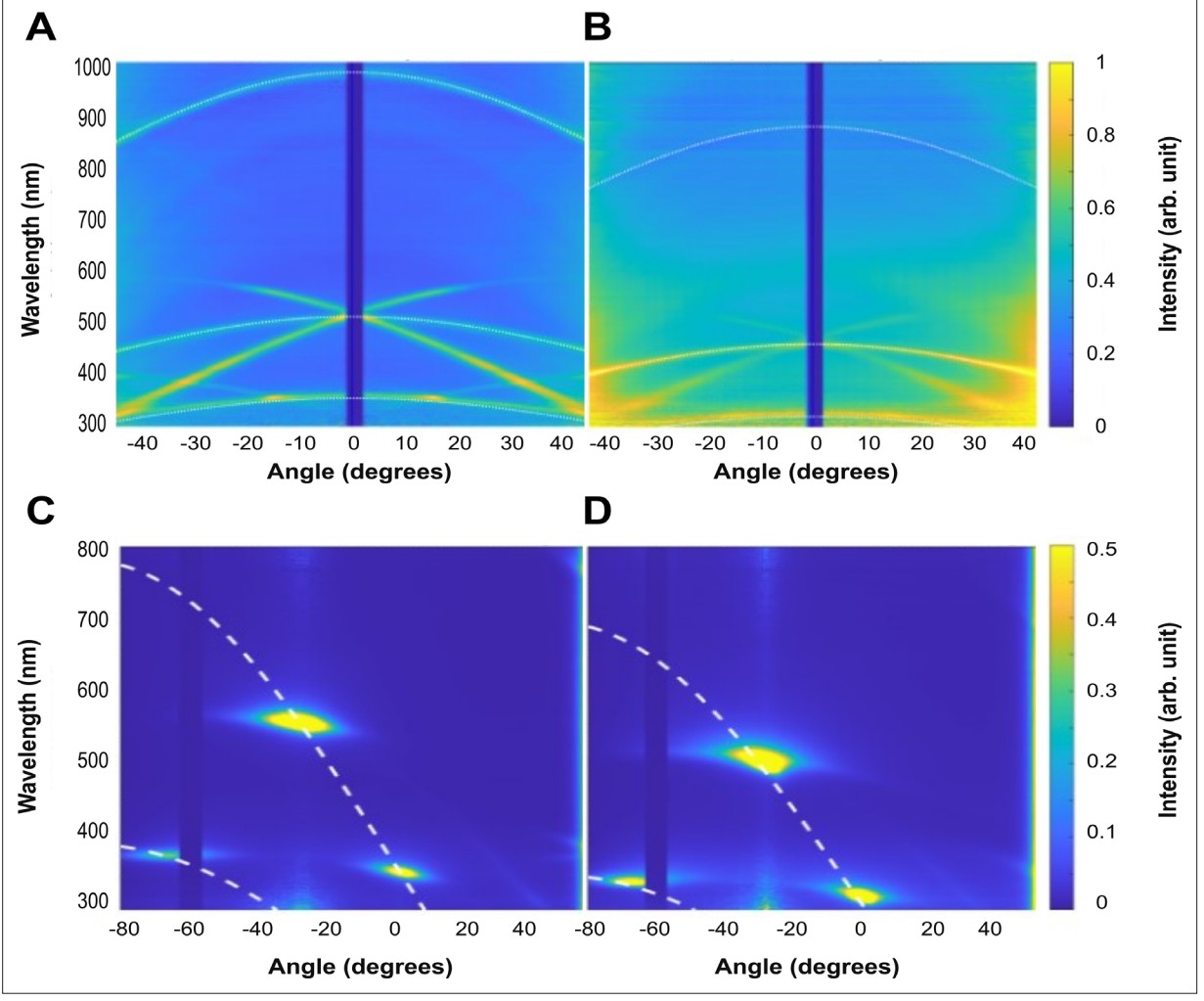

**Figure 5.** Goniometry analysis of IR1 WT and Δ*moe*A strains grown as a film layer on ASWBKC medium. Specular reflection analysis of (**A**) WT, and (**B**) Δ*moe*A, and scattering (light illumination with an angle of 60°) of (**C**) WT, and (**D**) Δ*moe*A. The dotted lines represent the values of the grating equation.

The online version of this article includes the following source data for figure 5:

**Source data 1.** Specular reflectivity, optical data for IR1 WT.

**Source data 2.** Optical scattering data for IR1 WT.

**Source data 3.** Specular data for the optical properties of colonies of IR1 Δ*moe*A.

**Source data 4.** Scattering data for Δ*moe*A.

involved (amino acid biosynthesis), and a nitrite reductase (nitrogen assimilation; *Supplementary file 1d*). An upregulated protein was NAD(P)H-nitrite reductase, also involved in nitrogen assimilation (*Supplementary file 1e*).

Of the 5471 known proteins in IR1, 58.1% (3,181 proteins) intracellular proteins were identified, 10.2% (324) showed significant differences (p<0.01), with 34.3% (111) considered downregulated, and 27.2% (88) upregulated in the Δ*moe*A (*Figure 7A*). The downregulated subset included 29 hypothetical proteins, while the upregulated subset had 31.

Downregulated intracellular proteins were involved in amino acid metabolism (10), RNA processing (10), transport (9), DNA transcription (8), translation (6), fatty acid metabolism (4), antimicrobial resistance (4), nucleotide metabolism (4), cofactor biosynthesis (4), proteolysis (4), biofilm formation (3), homeostasis (3), carbohydrate metabolism (3), and various metabolic processes (*Supplementary file 1d*). Additionally, 28 proteins with unknown functions were identified (*Supplementary file 1d*). Some downregulated proteins, such as an ABC transporter ATP-binding protein and a membrane assembly

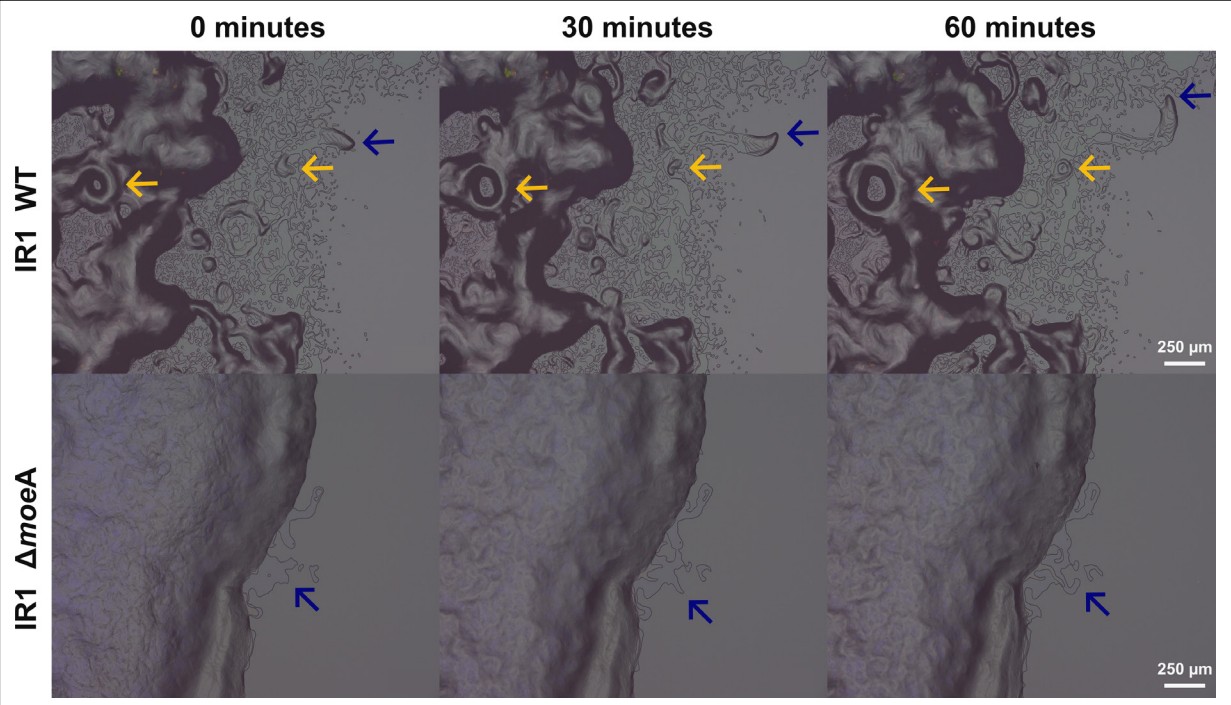

**Figure 6.** Images taken with a KEYENCE microscope using full coaxial light at the edge of the colony of IR1 WT and Δ*moe*A growing on ASWB. These are frames at 0 min, 30 min, and 60 min from the respective 1 hr time-lapse videos. The blue arrows indicated the motility of a group of cells, and the yellow arrows indicated the forming of circular 'vortex' patterning and movement.

protein (involved in phospholipid transformation), as well as an alanine dehydrogenase (amino acid metabolism), and some hypothetical proteins with unknown role, were completely repressed (found only in the WT).

Upregulated intracellular proteins were involved in transport (12), non-ribosomal peptide synthesis (11), stress response (6), carbohydrate metabolism (5), proteolysis (4), signaling (4), electron transport (3), glycosylation (3), DNA repair (3), cofactor biosynthesis (2), and various metabolic processes (*Supplementary file 1e*). Additionally, 20 proteins with unknown roles were identified (*Supplementary file 1e*). Notably, among the most upregulated proteins, we observed a 23 S rRNA (adenine(1618)-N(6))-methyltransferase (involved in RNA processing), a hypothetical protein (unknown role), a chalcone isomerase (stress response), an aminopeptidase (proteolysis), and a transcriptional regulator (regulation of DNA transcription).

Of the total known proteins in IR1, 27.5% (1504 proteins) proteins were detected in the extracellular fraction, 60.4% (909) were statistically significant (p<0.01), with 20.5% (186) considered downregulated, and 20% (182) upregulated in Δ*moe*A (*Figure 7B*). The downregulated subset included 44 hypothetical proteins, while the upregulated subset had 70. Although fewer proteins were identified in the extracellular space compared to the intracellular space, a higher proportion was statistically significant and differentially regulated.

Analysis of downregulated proteins using SecretomeP showed that 5.4% (10) were likely secreted through a non-classical way, lacking typical secretion sequence motifs in their N-terminus. Additionally, SignalP analysis revealed that 31.7% (59) had a putative signal peptide, suggesting they are Sec (general secretory pathway) substrates and likely to be secreted. The downregulated proteins likely to be secreted (69) included those involved in carbohydrate metabolism (7), transport (7), stress response (4), antibiotic resistance (3), lipopolysaccharide assembly (3), protein modification (3), motility (2), and several other functions (*Supplementary file 1f*). Additionally, 27 proteins with unknown roles were identified (*Supplementary file 1f*). Notably, among the most highly downregulated proteins included a flagellin biosynthesis protein (unknown role), probably misannotated as the pathways for flagella synthesis are absent in *Flavobacterium* IR1, a murein hydrolase activator (cell division), a hypothetical

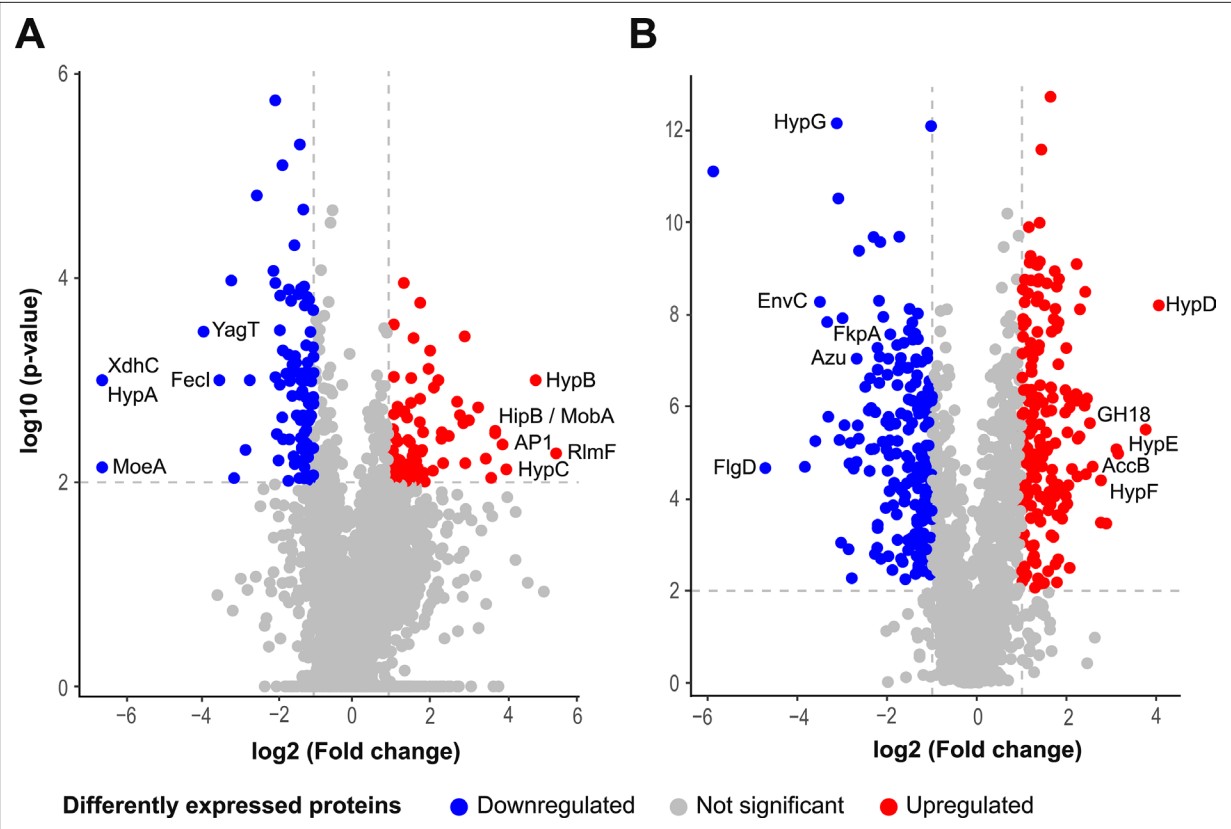

**Figure 7.** Volcano plots of the peptides identified in (**A**) the intracellular protein analysis, and in (**B**) the extracellular protein analysis. Some of the most regulated proteins are shown in the plots. The horizontal dashed lines represent the cut-off value for the p-value (2), and the vertical dashed lines represent the cut-off value for the fold change (–1 and 1).

The online version of this article includes the following source data for figure 7:

**Source data 1.** Cellular proteomics data for IR1 WT and Δ*moe*A.

**Source data 2.** Extracellular proteomics data for IR1 WT and ΔmoeA.

protein (lipopolysaccharide assembly), a peptidylprolyl isomerase (protein modification), and an azurin (electron transport).

Analysis of upregulated proteins using SecretomeP revealed that 6.0% (11) potentially follow a non-classical secretion pathway. SignalP analysis indicated that 54.4% (99) of the upregulated proteins possessed a signal peptide. The upregulated proteins likely to be secreted (111) included those involved in transport (19), carbohydrate metabolism (18), proteolysis (12), stress response (5), fatty acid metabolism (4), and other biological processes (*Supplementary file 1g*). Additionally, 45 proteins with unknown roles were identified (*Supplementary file 1g*). Notably, the most upregulated proteins included two hypothetical proteins (involved in unknown roles), a hypothetical protein (cell division), an acetyl-CoA carboxylase biotin carboxyl carrier protein (fatty acid biosynthesis), a glycoside hydrolase (carbohydrate metabolism), and a hypothetical protein (transport).

The combination of protein analysis and genomic data from the IR1 genome provided insights into the putative operons or gene clusters affected by the deletion of *moe*A (*Figure 8*). Intracellular proteomic analysis suggested the downregulation of putative operons associated with antimicrobial drug resistance, fatty acid biosynthesis, purine catabolism, and phospholipid transformation. Conversely, putative operons involved in respiratory electron transport, carbohydrate metabolism, non-ribosomal peptide synthesis, antioxidant stress, and cell wall synthesis were upregulated. In the extracellular proteomic analysis, a putative operon with an unknown function was downregulated, while putative operons involved in fatty acid biosynthesis, carbohydrate metabolism, and unknown functions were upregulated. Notably, the deletion of *moe*A created a cascade of regulation effects that affected pathways not previously linked to molybdopterin synthesis.

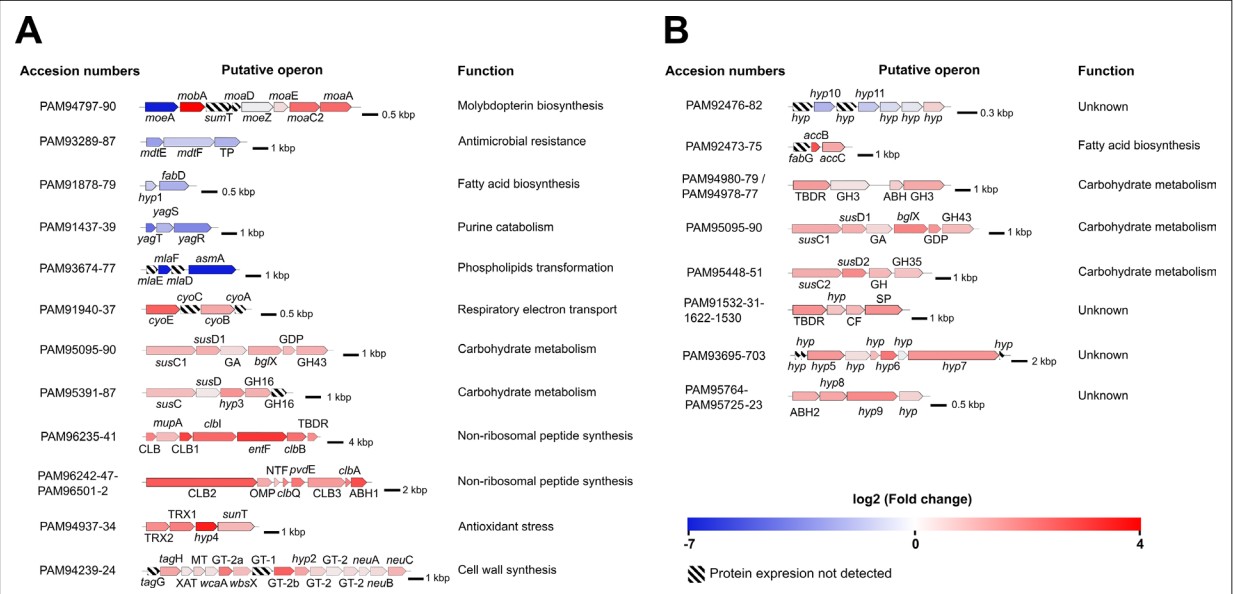

**Figure 8.** Putative operons or gene clusters with differentially expressed proteins identified in (**A**) intracellular and (**B**) extracellular proteomic analyses, based on function and proximity. To the left of each operon are the accession numbers of the translated proteins, to the right is the predicted function. Gene or protein names are indicated. Genes are colored based on the fold change of the encoded proteins. The black bars show the scale in kilobase pairs (kbp). TP: transporter, *hyp*: hypothetical protein, GA: glycoamylase, GHX: glycosyl hydrolase family X, GDP: glycerophosphoryl diester phosphodiesterase, CLB: colibactin biosynthesis, TBDR: TonB-dependent receptor, OMP: outer membrane protein, NTF: nuclear transport factor 2, ABH: alpha/beta hydrolase, TRX: thioredoxin domain-containing protein, XAT: xenobiotic acyltransferase, MT: SAM-dependent methyltransferase, GT-X: glycosyltransferase family X, CF: cell surface protein, SP: secretion protein.

Previous studies, alongside the results of this investigation, have shown the importance of complex polysaccharides degradation in the development of SC (*Johansen et al., 2018*; *van de Kerkhof et al., 2022*). In the Bacteroidetes phylum, polysaccharides utilization loci (PUL) operons facilitate the uptake and processing of these polysaccharides. Typically, PUL operons consist of a tandem pair of genes resembling *sus*CD, which encode a transport and substrate-binding complex, and various carbohydrate active enzymes (CAZymes), such as glycosyl hydrolases and pectate lyases (*Terrapon et al., 2015*).

Our intracellular and extracellular protein analysis revealed the upregulation of three putative PUL operons with similar organization (*Figure 8*): (1) PAM95095-90, which includes a glycoamilase, a glycosyl hydrolase family 3 (GH3) involved in cellulose degradation, a glycerophosphoryl diester phosphodiesterase, and a GH43 that degrades hemicellulose and pectin polymers *Ara et al., 2020*; *Mewis et al., 2016*; (2) PAM95448-51, which includes an unidentified GH, and a GH35 enzyme that hydrolyzes terminal non-reducing β-D-galactose residues *Tanthanuch et al., 2008*; (3) PAM95391-88, which includes a hypothetical protein, and two GH16, one of which was not detected, involved in the degradation of various polysaccharides such as agar and kappa-carrageenan (*Viborg et al., 2019*). Additionally, other carbohydrate metabolism-related proteins were upregulated in the Δ*moe*A, including a GH18 enzyme involved in chitin degradation (*Chen et al., 2020*), and a pectate lyase involved in starch degradation (*Supplementary file 1g*; *Aspeborg et al., 2012*).

### *moe*A deletion affects metabolism of complex carbohydrates

As previously described, the IR1 WT and Δ*moe*A strain were grown on various complex polysaccharides, showing different color phenotypes. The Δ*moe*A colony displayed a strong blue SC phenotype on ASWBKC, a dull green on ASWBF, and a dull blue center with a blue internal ring and green external ring on ASWBS (*Figure 9*). These results suggest a connection between SC, moeA, and polysaccharide metabolism. Proteins linked to carbohydrate metabolism were also highly regulated, reinforcing this link (*Supplementary file 1f and g*). Both strains were grown on ASWS, and starch degradation was visualized using iodine vapor (*Kasana et al., 2008*). The colonies were photographed from the front and the back (*Figure 9*). The WT strain showed a duller and smaller starch degradation zone

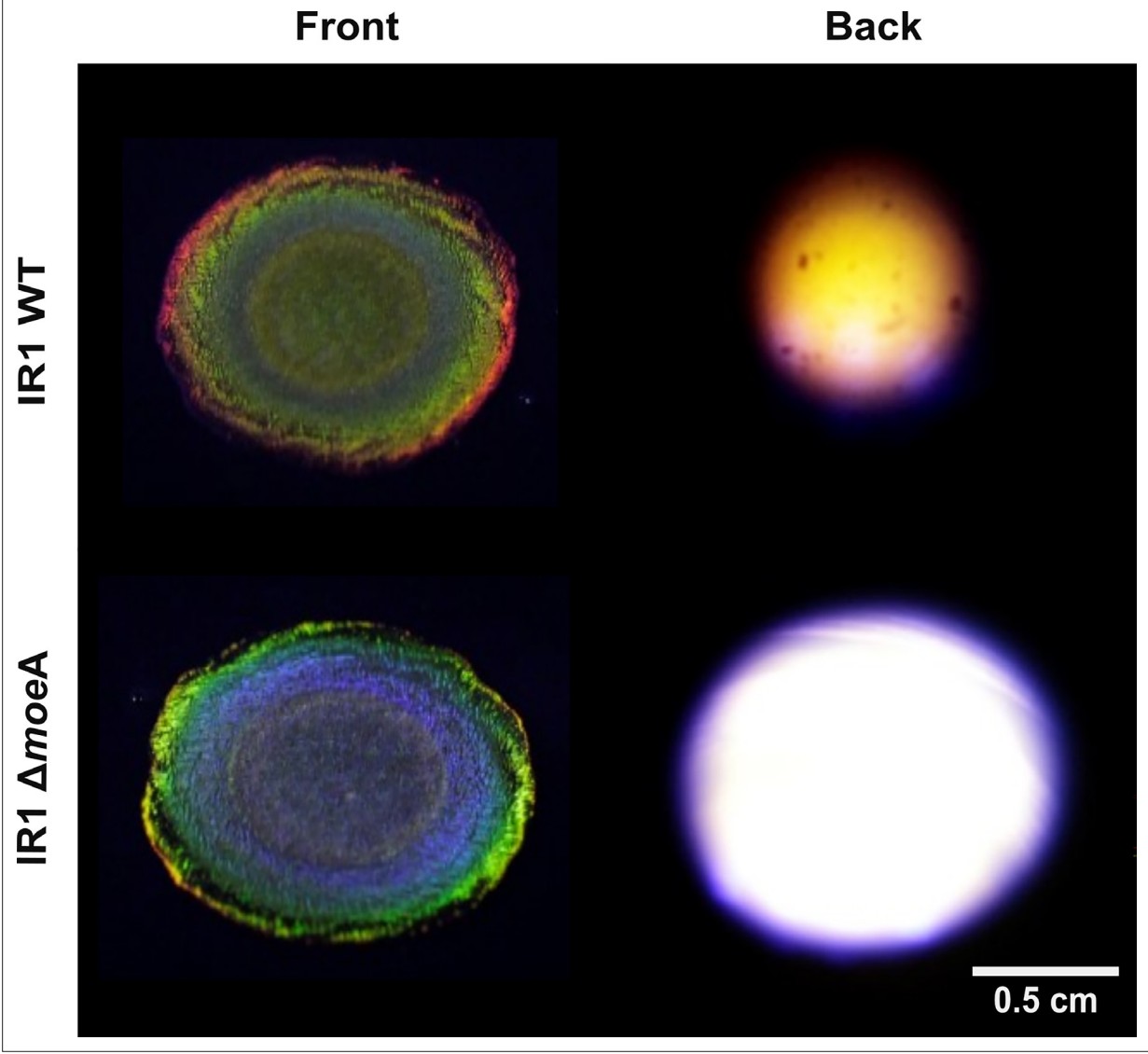

**Figure 9.** Colonies of IR1 WT (top row) and ∆*moe*A (bottom) grown on ASWS. Iodine vapor was used to dye the starch remaining in the media. The zones of starch degradation are seen as the lighter areas under the colonies. The images were taken at the same 90° angle from the front (left column) and back (right) of the plate.

(0.58±0.12 cm) compared to ∆*moe*A (1.17±0.17 cm). In contrast to other media where ∆*moe*A colony expansion was less than WT, the ∆*moe*A showed similar colony spreading and stronger starch degradation, supporting a role of *moe*A in complex polysaccharides metabolism.

## Discussion

SC in biological systems is well-studied optically, but less well understood genetically. This study aimed to expand and deepen the knowledge of genes involved in bacterial SC, focusing on the predicted SC-related gene, *moe*A (*Zomer et al., 2024*). By deleting *moe*A from the IR1 genome, a model for bacterial SC, we conducted microbiological, optical, proteomic, and comparative genomic analyses of the mutant. The results demonstrated the possibility of engineering SC by targeting specific pathways.

The *moe*A gene is a part of the molybdenum cofactor (MoCo) synthesis pathway, which is not exclusive to bacteria, but also found in archaea, animals, and plants, tracing back to the last universal common ancestor (*Allen et al., 1994*; *Weiss et al., 2016*). MoCo is essential for molybdoenzymes that

catalyze oxo-transfer and hydroxylation reactions such as nitrate reductase, xanthine dehydrogenase, and aldehyde oxidase (*Wootton et al., 1991*; *Zhang and Gladyshev, 2008*). In the ΔmoeA proteome, the absence of MoeA produced the downregulation of some molybdoenzymes like xanthine hydrogenases, aldehyde oxidase, and nitrite reductase, suggesting their synthesis depends on MoCo availability. However, one nitrite reductase protein (PAM94801) was upregulated, potentially independent of MoCo. Additionally, proteins from *moaA*, *moaC2*, and *mobA* genes which are present in the same operon as *moeA* were upregulated, possibly to boost molybdopterin availability for MoCo synthesis.

The presence of *moeA* in the genome or the putative operon structure for MoCo pathway alone does not determine a bacterial strain's ability to form SC colonies. For example, the genomes of the Bacteroidetes strains *Flavobacterium* IR1, *F. johnsoniae* UW101, and *Cellulophaga lytica* HI1, which contain all the genes for the synthesis of MoCo, display SC. Meanwhile, *M. algicola* HM30 and *Z. galactinovorans* DSM1208, lacking *moaD* and *moaC*, respectively, also show SC. Interestingly, the corresponding proteins of *moaD* and *sumT* were not detected in the proteomic analysis of IR1 WT and ΔmoeA. Additionally, *moaD* and *mobA* were not present in all SC strains. Thus, we concluded that the presence of *moaC*, *moaD*, *mobA*, and *sumT* is not essential for SC formation (*Zomer et al., 2024*).

The predominantly green SC of IR1 WT has been studied using transposon mutagenesis, cultivation, and optical characterization, revealing additional SCs like yellow, orange, red, blue, and purple (*Johansen et al., 2018*; *van de Kerkhof et al., 2022*). Here, *moeA* was deleted from the IR1 genome using SIBR-Cas (*Patinios et al., 2021*), resulting in a strong blue shift in the colony color, confirmed and quantified by goniometry. The WT and ΔmoeA colonies show variations in color, color pattern, and intensity depending on three conditions: (1) observation angle, displaying green, yellow, and blue hues with different intensities; (2) the presence of peptone and yeast extract, affecting color and motility; and (3) the type of polysaccharides present in the media, which significantly altered color and motility. These findings showed that SC color hue, pattern, and intensity can be modified by genetic engineering, observation angle, and nutrient changes.

Previously, mutations in *trmD* tRNA methyltransferase and a *clbB* triosephosphate isomerase were described (*Johansen et al., 2018*). A transposon insertion in *trmD* led to the loss of SC, while preserving growth and motility, and a *clbB* disruption produced a dull green/blue SC. Here, TrmD was downregulated, and ClbB was upregulated in the ΔmoeA proteomics analysis. Deleting *moeA* also caused downregulation of GldL, a protein essential for gliding motility and secretion in *F. johnsoniae* (*Shrivastava et al., 2013*). The reduced motility in the ΔmoeA mutant may have resulted from the combined downregulation of *trmD*, GldL, ribosomal proteins, and other uncharacterized proteins. Additionally, the upregulation of ClbB and other regulated proteins may contribute to the SC shift from green to blue.

Polysaccharide metabolism in IR1 has been linked to changes in colony color and motility through the study of fucoidan metabolism (*van de Kerkhof et al., 2022*). Polysaccharide degradation and gliding motility are coupled to the same mechanism: the phylum-specific type IX secretion system, used for the secretion of enzymes and proteins involved in both functions (*McKee et al., 2021*). Although *moeA* has not been previously linked to polysaccharide degradation (*Hasona et al., 1998*; *Tao et al., 2005*; *Leimkühler, 2017*), its deletion led to the upregulation of proteins from three PUL operons and others involved in polysaccharide metabolism, likely causing the color shift from green (WT) to blue (ΔmoeA). The identified proteins were involved in degrading cellulose, hemicellulose, pectin, galactose polymers, agar, kappa-carrageenan, mannanose, and starch. The polysaccharide degradation versatility was supported by checking the starch degradation in both strains, with the ΔmoeA capable of degrading starch faster and more efficiently than WT, producing larger and clearer halos with iodine staining.

On different polysaccharide media, the ΔmoeA strain showed varied SC and colony expansion patterns: green/blue SC and low colony expansion on agar, intense blue SC and low colony expansion on kappa-carrageenan, dull green SC and low colony expansion on fucoidan, and blue/green SC with higher colony expansion on starch. Interestingly, the color phenotype of the WT and ΔmoeA exchanged their phenotype on kappa-carrageenan (a simple linear sulfated polysaccharide of D-galactopyranose) and fucoidan (a complex sulfated polysaccharide of fucose and other sugars as galactose, xylose, arabinose, and rhamnose), showing the importance of the polysaccharide metabolism in SC. While reduced motility has been associated with dull or absent SC and reduced polysaccharide metabolism (*Kientz et al., 2012a*; *Johansen et al., 2018*), ΔmoeA showed reduced motility, but an

intense blue SC and high polysaccharide metabolism. Based on these results, we established a link among polysaccharide metabolism, MoCo biosynthesis, and SC, showing that intense SC is not strictly dependent on motility.

Ecologically, we hypothesize that dense, highly structured bacterial colonies, such as necessary for the SC phenotype, can enhance the uptake of metabolic degradation products from complex polysaccharides. These large macromolecules are often partially hydrolyzed extracellularly because they are too large to pass through bacterial cell membranes. For example, marine Vibrionaceae strains that produce lower levels of extracellular alginate lyases tend to aggregate more strongly, potentially facilitating localized degradation and uptake of polysaccharides (*D'Souza et al., 2023*). Additionally, certain marine bacteria employ a 'selfish' mechanism to internalize large polysaccharide fragments into their periplasmic space, minimizing loss to the environment and enhancing substrate utilization (*Reintjes et al., 2017*). Bacteria secrete enzymes into the surrounding environment to break these polysaccharides down into more easily absorbable monosaccharides or oligosaccharides. This mechanism suggests that the colony structure could create a physical barrier that keeps these products concentrated and near the cells, allowing the colony to efficiently access and utilize these products, preventing the leakage into the surrounding environment. While SC may also yield other ecological benefits associated with growth in biofilms, the highly structured colonies that characterize SC may be more resistant against invasion by competitor species scavenging for degradation products than an unstructured biofilm. This model is consistent with the observation that SC is associated with polysaccharide metabolism genes, and with the recent observation that SC is mainly localized on surface and interface environments such as air-water interfaces, tidal flats, and marine particles (*Zomer et al., 2024*).

SC bacteria like *C. lytica* (*Sullivan et al., 2023*) and *Flavobacterium* IR1 *Groutars et al., 2022* have been recently studied to be used as colorful biomaterials, making genetic engineering to modify SC a potential next step for developing new colorants. Similar to IR1, *C. lytica* belongs to the *Flavobacteriaceae* family, exhibiting gliding motility, similar SC, and has diverse polysaccharide metabolism genes, though it lacks genetic engineering tools (*Kientz et al., 2012a*; *Kientz et al., 2016*; *Lisov et al., 2022*). Genetic engineering SC in IR1 opens the way to synthetic biology of SC and its application in biomaterials, offering a sustainable alternative to traditional pigments.

In conclusion, our results demonstrate the capacity to engineer bacterial SC based on a prediction provided by genomics. The simple deletion of one gene, *moe*A, shifted the SC of IR1 colony from green to blue, while nutrient and polysaccharide availability emerged as key factors affecting SC color and motility. Proteomics analysis revealed polysaccharide metabolism as a driver of SC changes, hinting at a possible ecological significance. Additionally, several uncharacterized proteins were differentially expressed in the *moe*A KO, providing exciting new leads for further exploration of bacterial SC. This study marks a step forward in the synthetic biology of SC, with promising applications in biomaterials.

## Materials and methods
### Bioinformatics analysis of the molybdopterin pathway operon in *Flavobacterium* IR1

Synteny and homology of the proteins related to SC were visualized with gggenomes 1.0.0 (*Hackl et al., 2024*) in RStudio 1.1.456. First, sequences of genomes and SC proteins were obtained from a previous work (*Zomer et al., 2024*). Proteins were predicted in the genomes with Prodigal 2.6.3 (*Hyatt et al., 2010*). Proteins of interest were matched with BLAST 2.14.0+blastp (*Altschul et al., 1990*) against Prodigal's predicted proteins to find genomic coordinates. Operon start coordinate matches the start of the first gene of the putative operon and operon end coordinate matches the end of the last gene of the putative operon. Python 3.12.4 and Jupyter Notebook 7.2.1 were used to adapt file formats and create objects compatible with gggenomes. The corresponding phylogenetic tree was made from aligned 16 S rRNA genes using Barrnap 0.9 (*Seemann, 2024*), BEDtools 2.31.0 getfasta (*Quinlan and Hall, 2010*), MAFFT 7.505 (*Katoh and Standley, 2013*), iqtree v1.6.2 (*Minh et al., 2020*), and iToL online v6 (*Letunic and Bork, 2024*). The final figure containing synteny, homology, and the tree was done in Inkscape 1.3.2. The tutorial and scripts for reproducing the figure were stored in a GitHub repository: https://github.com/MGXlab/genes_synteny (copy archived

at *MGXlab, 2026*). Tools were used with their default parameters and exact commands can be found in the GitHub repository.

## Bacterial strains and growth conditions

Bacterial strains used in this study are described in *Supplementary file 1a*. *Flavobacterium* iridescence species 1 (IR1) was the target strain used in this project. IR1 was grown in Artificial Sea Water (ASW) medium composed of 5 g·L⁻¹ peptone (Sigma-Aldrich), 1 g·L⁻¹ yeast extract (Sigma-Aldrich), and 10 g·L⁻¹ sea salt (Lima), at 25 °C and grown in an orbital incubator at 200 rpm (*Johansen et al., 2018*). *Escherichia coli* DH5α (New England Biolab, NEB) was used for general plasmid propagation and standard molecular techniques. *E. coli* was grown in Luria-Bertani (LB) medium composed of 10 g·L⁻¹ tryptone (Sigma-Aldrich), 5 g·L⁻¹ yeast extract, and 10 g·L⁻¹ NaCl (Sigma-Aldrich), at 37 °C shaken at 200 rpm. IR1 was plated on ASW with 1% agar (Invitrogen) with or without 0.25 g·L⁻¹ nigrosine (Sigma-Aldrich) (*Johansen et al., 2018*). *E. coli* was plated on LB medium containing 1.5% agar (Invitrogen). Media were supplemented with 50 µg·mL⁻¹ spectinomycin (Sigma-Aldrich), 100 µg·mL⁻¹ ampicillin or 200 µg·mL⁻¹ erythromycin (Sigma-Aldrich) when necessary. All the strains were stored in 25% glycerol solution at –80 °C.

## Plasmid construction

All the plasmids used for SIBR-based gene knockout (KO) were constructed from pSIBR048 (*Supplementary file 1b*) following the previously described protocol by Patinios and coworkers (*Patinios et al., 2021*). In brief, to introduce the *moe*A homologous arms (HA) and mediate the deletion of *moe*A, pSIBR048 was linearized using MluI (NEB) and the phosphorylated ends were removed using Shrimp Alkaline Phosphatase (NEB). 1500 bp HA corresponding upstream and downstream of *moe*A were amplified from the IR1 genome by PCR with Dream Taq DNA Polymerase (Thermo Fisher). The amplicons were resolved on 1% agarose (Eurogentec) electrophoresis gel and purified using GenElute PCR Clean-Up Kit (Sigma-Aldrich). The PCR products were introduced to the linearized pSIBR048 using NEBuilder HiFi DNA Assembly Master Mix (NEB), resulting in the pMoeA_NT. Following this, the *moe*A targeting spacer was introduced in the pMoeA_S1 plasmid as previously described (*Patinios et al., 2021*). The DNA sequence of each newly created plasmid was verified by Sanger sequencing. Oligonucleotides used in this study are listed in *Supplementary file 1c*.

## *E. coli* DH5α competent cell preparation and transformation

Competent cells of *E. coli* DH5α, for chemical transformation, were prepared following the CaCl₂ method described by Sambrook (*Sambrook et al., 1989*). The cells were aliquoted ready to be used or stored at –80 °C. The transformation of the competent DH5α cells was done by heat shock following the High Efficiency Transformation Protocol of NEB. For this protocol, LB medium was used instead of SOC medium. The cells were plated on LB 1.5% agar supplemented with 50 µg·mL⁻¹ spectinomycin and incubated at 37 °C for 1 day.

## IR1 competent cell preparation, transformation, and SIBR-Cas genetic engineering assay

The methods used for the preparation of the electro-competent cells of IR1, transformation with plasmids and SIBR-Cas genetic engineering were as previously described (*Patinios et al., 2021*). Mutant colonies were identified through colony PCR using primers cFwd *moe*A and cRev *moe*A, and Sanger sequencing (Eurofins).

## Effects of nutrient composition on SC in IR1 WT and *Δmoe*A

The visual phenotype of the mutant in comparison to the WT was first checked on agar plates under different nutrient conditions. ASWB agar contains ASW medium with 1% agar and 0.25 g·L⁻¹ nigrosine (*Johansen et al., 2018*). ASWB low nutrient medium (ASWBLow) contains the same nutrients as ASWB but without peptone (*Johansen et al., 2018*). Minimal medium (MM) contains 0.5% sea salt, 0.1% MgSO₄, 0.25% kappa-carrageenan (Special Ingredients) and 1% agar. ASWB kappa-carrageenan (ASWBKC) contains the same nutrients as ASWB, but with kappa-carrageenan instead of agar (ASWBC modified from *Johansen et al., 2018*). ASWB fucoidan (ASWBF) contains the same nutrients as ASWB plus 1% fucoidan (Absonutrix) (*Johansen et al., 2018*). ASWB starch (ASWBS) contains the same

nutrients as ASWB plus 1% starch (Sigma-Aldrich) (*Johansen et al., 2018*). Before studying the effects of the nutrient composition, both strains were cultivated overnight at 25 °C on an ASWB plate from which some bacterial biomass was collected, resuspended in 1% sea salt, and 10 µL of the bacteria suspension was spotted on the plates. The mutant was observed after 2 days by eye to check the display of SC.

## Imaging

Photographs of colonies were taken with a Canon digital camera equipped with a RF 100 mm macro lens or using a KEYENCE VHX-7000 Digital Microscope using defined angles of illumination and data capture (*Figure 2B*).

## Determining colony spread

Colonies of IR1 WT and Δ*moeA* were grown as a spot on ASWB, ASWBKC, ASWBF, ASWBS, ASWBLow, and MM for 6 days at 25 °C. The diameter of the colonies was measured at two time points, just after the spot was inoculated and after 6 days. These data were measured in triplicates for each condition and strain.

## Angle-resolved spectroscopy (goniometry)

The optical properties of the bacteria colonies were studied following the method previously described (*Johansen et al., 2018*). Angle-dependent reflectance spectra were measured using a custom-built goniometer setup (*Vignolini et al., 2013*) both in scattering and specular configuration. The samples were illuminated from a fixed direction by a Xenon lamp (Ocean Optics HPX-2000), and the reflected light was collected at different detection angles (resolution 1°) using a rotating arm connected to a spectrometer (Avantes HS2048) via an optical fiber. Data presented in this work were normalized against a white diffuser (Labsphere SRS-99–010).

## Analysis of the optical response

Angle-resolved reflectance spectra show peculiar features caused by the two-dimensional structural organization. In scattering configuration, diffraction spots are visible that can be correlated to the diffraction grating formed by the bacteria on the surface (*Schertel et al., 2020*; *Johansen et al., 2018*). More specifically, the angles of constructive interference from a diffraction grating can be expressed by the grating equation: $\theta_m = \arcsin(m\lambda/d - \sin\theta_i)$, where $m \in [0,-1,+1,-2,+2, …]$ is the diffraction order, $\lambda$ is the wavelength of light, d is the period of the structure, $\theta_i$ is the angle of incidence and $\theta_m$ is the reflection angle for a given order. This equation can be used to determine the period (d) of the bacteria organization, and deviation from the predicted diffraction spots can quantitatively inform about the degree of disorder compared to an ideal periodic structure. Information on the effective refractive index can be obtained from goniometry data acquired in specular configuration. In this case, reflectance peaks arise from the constructive interference of light with the multilayer structure and depend on various parameters. Considering both Bragg's law and Snell's law, the peak reflection wavelength $\lambda_B$ and corresponding incident angles $\theta_{in}$ at which constructive interference occur are linked via the following equation: $\lambda_B = 2n_{avg}\cdot d\cdot\cos(\arcsin(\sin\theta_{in}/n_{avg}))$, where $\theta_{in}$ is the illumination angle and $n_{avg}$ is the volume average effective refractive index of the total material composite in the photonic crystal. For construction, the angle of observation $\theta_{out}$ equals $\theta_{in}$.

## Intracellular and extracellular proteome sample preparation

WT IR1 and the *moe*A mutant were selected for intracellular and extracellular proteomics analysis. Cells were grown for 2 days at 25 °C completely covering ASWBKC plates. To prepare the whole cell fractions, cultures were harvested and centrifuged at 12,000 rpm for 15 min at 4 °C in 2 ml tubes. Cells were washed with 1% KCl solution, centrifuged at 12,000 rpm for 15 min at 4 °C and cell pellets were stored at –80 °C. For preparation of extracellular protein fractions, supernatants were collected after the first cell centrifugation, the supernatants were transferred into new 2 ml tubes, and centrifuged at 12,000 rpm for 25 min at 4 °C. To ensure reproducibility, both preparations were performed in biological triplicates.

Peptides originating from IR1 intracellular and extracellular proteins were extracted according to the protocol described by Campos and coworkers (*Campos et al., 2015*; *Campos et al., 2016*). The

resulting dried peptides were resuspended in 0.1% formic acid in deionized water followed by bath-sonication for 5 min and 5 min centrifugation at 12,000 rpm at 25 °C. Peptide concentration was assessed at A280 using ND-1000 Nanodrop spectrophotometer (Thermo Scientific) peptide concentrations were adjusted to 0.1 mg/ml to normalize samples prior to LC-MS/MS analyses.

## Proteome sample analysis

For the LC-MS/MS analyses, peptides were separated by EASY-nLC II system (Thermo Scientific) at flow rate of 300 nl/min on a precolumn (Acclaim PepMap 100, 75 µm×2 cm, Thermo Fisher Scientific) followed by EASY-Spray C18 reversed-phase nano LC column (PepMap RSLC C18, 2 µm, 100 A 75 µm×25 cm, Thermo Fisher Scientific) thermostated at 55 °C. A 90 min gradient of 0.1% formic acid in water (A) and 0.1% formic acid in 80% acetonitrile (B) was distributed as follows: from 6% B to 30% B in 65 min; from 30% B to 100% B in 20 min and hold at 100% B for 5 min. Automated online analyses were performed in positive ionization mode by a Q Exactive HF mass spectrometer (Thermo Fisher Scientific) equipped with a nano-electrospray. Full scans were performed at resolution 120,000 in a range of 380–1400 m/z and the top 15 most intense multiple charged ions were isolated (1.2 m/z isolation window) and fragmented at a resolution of 30,000 with a dynamic exclusion of 30 s. The generated raw files were analyzed using Sequest HT in Proteome Discoverer software (Thermo Fisher Scientific, San Jose, CA, USA, CS version 2.5.0.400). *Flavobacterium* (NCBI Taxonomy ID 2026304) protein sequence database used for protein identification was acquired from NCBI (https://www.ncbi.nlm.nih.gov/); downloaded on 10th of February 2023; 5468 entries. The following search parameters were used: trypsin as a digestion enzyme; maximum number of missed cleavages 2; fragment ion mass tolerance 0.08 Da; parent ion mass tolerance 10 ppm; carbamidomethylation of cysteine as fixed modification and methionine oxidation as variable modifications.

## Proteome bioinformatics

Scaffold (version Scaffold_5.3.0, Proteome Software Inc, Portland, OR) was used to validate protein identifications and for relative quantification of proteins. Peptide identifications were accepted if they could be established at greater than 90% probability by the Scaffold Local FDR algorithm. Protein identifications were considered correct if they could be established at a greater than 95% probability and contained at least 1 unambiguously identified peptide. Protein probabilities were assigned by the Protein Prophet algorithm (*Nesvizhskii et al., 2003*). Proteins that contained similar peptides and could not be differentiated based on MS/MS analysis alone were grouped to satisfy the principles of parsimony. Proteins sharing significant peptide evidence were grouped into clusters. These clusters were associated with a specific protein of IR1 within the GenBank database, giving the following default identity name: PAM9XXXX.

## Proteome data analysis

The quantitative protein abundance levels were analyzed in the proteins that had a difference between the sample groups when applying the Student's t-test, using the multiple test correction Benjamini-Hochberg, and a cut-off p-value lower than 0.01 was chosen for statistically significant quantitative difference in relative proteins amount between *moe*A and WT sample groups. A protein was considered downregulated when the log2 of the fold change (Δ*moe*A/WT) was lower than –1, and upregulated when it was higher than 1.

The identified differentially expressed proteins were bioinformatically analyzed using the KEGG tool BlastKOALA for functional characterization and the InterProScan software (*Kanehisa et al., 2016*; *Jones et al., 2014*). The proteins identified in extracellular fractions were also analyzed using SecretomeP (identifies signal-independent secreted proteins) and SignalP (predict signal peptides) software to confirm that they were predicted to be potentially secreted and to exclude possible contamination by intracellular proteins (*Bendtsen et al., 2005*; *Teufel et al., 2022*).

## Monitoring starch degradation by iodine staining assay

Colonies of IR1 WT and Δ*moeA* were grown on plates with ASWS (ASWBS without nigrosin) for 2 days at 25 °C. Iodine crystals were deposited on the lid of the plate and incubated upside down overnight to expose the agar to the iodine vapor. The plates were checked for starch degradation which

corresponds to the zones of clearing, with dark, stained areas indicating presence of undegraded starch (*Kasana et al., 2008*). These were measured in triplicates for each condition and strain.

## Acknowledgements

This project is supported by the European Union's Horizon 2020 research and innovation program under Marie Skłodowska-Curie grant agreement No 860125 (SV, CJI, AED, and MM), the European Research Council (ERC) Consolidator grant 865694: DiversiPHI, the Deutsche Forschungsgemeinschaft (DFG, German Research Foundation) under Germany's Excellence Strategy – EXC 2051 – Project-ID 390713860, the Alexander von Humboldt Foundation in the context of an Alexander von Humboldt-Professorship founded by the German Federal Ministry of Education and Research, VIDI grant (VI.Vidi.203.074) from The Netherlands Organization for Scientific Research (NWO) (RHJS), and B-INK Proof of Concept grant from the ERC 101188114 (SV). Open access funding provided by Max Planck Society.

## Additional information

### Funding

| Funder | Grant reference number | Author |
| --- | --- | --- |
| Horizon 2020 Framework Programme | 10.3030/860125 | Álvaro Escobar Doncel Maria Murace Silvia Vignolini Colin Ingham |
| European Research Council | 10.3030/865694 | Bas E Dutilh |
| Deutsche Forschungsgemeinschaft | 390713860 | Bas E Dutilh |
| Nederlandse Organisatie voor Wetenschappelijk Onderzoek | | Raymond HJ Staals |
| European Research Council | 10.3030/101188114 | Silvia Vignolini |
| Alexander von Humboldt Foundation | | Bas E Dutilh |

The funders had no role in study design, data collection and interpretation, or the decision to submit the work for publication. Open access funding provided by Max Planck Society.

### Author contributions

Álvaro Escobar Doncel, Data curation, Formal analysis, Investigation, Visualization, Methodology, Writing - original draft, Project administration, Writing – review and editing; Constantinos Patinios, Resources, Data curation, Writing – review and editing; Alexandre Campos, Maria V Turkina, Formal analysis, Writing – review and editing; Maria Beatriz Walter Costa, Software, Formal analysis, Writing – review and editing; Maria Murace, Formal analysis; Raymond HJ Staals, Resources, Supervision, Writing – review and editing; Silvia Vignolini, Supervision, Funding acquisition, Project administration, Writing – review and editing; Bas E Dutilh, Conceptualization, Supervision, Project administration, Writing – review and editing; Colin J Ingham, Conceptualization, Resources, Data curation, Supervision, Funding acquisition, Project administration, Writing – review and editing

### Author ORCIDs
Álvaro Escobar Doncel https://orcid.org/0000-0003-2836-5055
Colin J Ingham https://orcid.org/0000-0003-1508-019X

Reviewer #1 (Public review): https://doi.org/10.7554/eLife.105029.3.sa1

Reviewer #2 (Public review): https://doi.org/10.7554/eLife.105029.3.sa2
Author response https://doi.org/10.7554/eLife.105029.3.sa3

## Additional files

### Supplementary files

Supplementary file 1. Supplementary tables. a. Bacterial strains used in this study. b. Plasmids used in this study. c. Oligonucleotides used in this study. d. The most downregulated intracellular proteins in the Δ*moe*A mutant, and proteins mentioned in the main text and in *Figure 8*. * Gene name or protein with the designation that gives the most information. "-" means no gene name; *hyp*: hypothetical; TP: transporters. e. The 5 most upregulated intracellular proteins in the Δ*moe*A mutant, and proteins mentioned in the main text and in *Figure 8*. * Gene name or protein with the designation that gives the most information. "-" means no gene name; *hyp*: hypothetical; GT-2: glycosyltransferase family 2; GHX: glycosyl hydrolase family X; TRX: thioredoxin domain-containing protein; TBDR: TonB-dependent receptor; OMP: outer membrane protein; CLB: colibactin biosynthesis ABH: alpha/beta hydrolase. f. The most 5 downregulated extracellular proteins in the Δ*moe*A mutant, and proteins mentioned in the main text and in *Figure 8*. * Gene name or protein with the designation that gives the most information. "-" means no gene name; *hyp*: hypothetical. ** SP: signal peptide; NC: non-classical; -: not secreted. g. The 5 most upregulated extracellular proteins in the Δ*moe*A mutant, and proteins mentioned in the main text and in *Figure 8*. * Gene name or protein with the designation that gives the most information. ABH: alpha/beta hydrolase; SP: secreted protein; CF: cell surface. ** SP: signal peptide; NC: non-classical; -: not secreted.

MDAR checklist

### Data availability

All data generated or analysed during this study are included in the manuscript and supporting files; source data files have been provided.

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
