## [Editor Report · eLife Assessment]

This manuscript presents **important** findings on how structural color can be manipulated through a specific single-gene mutation in a motile bacterium. **Compelling** data provide a promising model to identify genes and molecular mechanisms supporting this widespread optical phenomenon. This work will be of interest to biophysicists and microbiologists working on structural colors and Flavobacterium.

---

## [Referee Report · Reviewer #1 (Public review)]

Structural colors (SC) are based on nanostructures reflecting and scattering light and producing optical wave interference. All kinds of living organisms exhibit SC. However, understanding the molecular mechanisms and genes involved may be complicated due to the complexity of these organisms. Hence, bacteria that exhibit SC in colonies, such as Flavobacterium IR1, can be good models.

Based on previous genomic mining and co-occurrence with SC in flavobacterial strains, this article focuses on the role of a specific gene, moeA, in SC of Flavobacterium IR1 strain colonies on an agar plate. moeA is involved in the synthesis of the molybdenum cofactor, which is necessary for the activity of key metabolic enzymes in diverse pathways.

The authors clearly showed that the absence of moeA shifts SC properties in a way that depends on the nutritional conditions. They further bring evidence that this effect was related to several properties of the colony, all impacted by the moeA mutant: cell-cell organization, cell motility and colony spreading, and metabolism of complex carbohydrates. Hence, by linking SC to a single gene in appearance, this work points to cellular organization (as a result of cell-cell arrangement and motility) and metabolism of polysaccharides as key factors for SC in a gliding bacterium. This may prove useful for designing molecular strategies to control SC in bacterial-based biomaterials.

---

## [Referee Report · Reviewer #2 (Public review)]

The authors constructed an in-frame deletion of moeA gene, which is involved in molybdopterin cofactor (MoCo) biosynthesis, and investigated its role in structural colors in Flavobacterium IR1. The deletion of moeA shifted colony color from green to blue, reduced colony spreading, and increased starch degradation, which was attributed to the upregulation of various proteins in polysaccharide utilization loci. This study lays the ground for developing new colorants by modifying genes involved in structural colors.

Overall, this is a well-written paper in which the authors effectively address their research questions through proper experimentation. This work will help us understand the genetic basis of structural colors in Flavobacterium and open new avenues to study the roles of additional genes and proteins in structural colors.

---

## [Author Response]

The following is the authors’ response to the original reviews.

**Public Reviews:**

**Reviewer #1 (Public review):**
Summary:Structural colors (SC) are based on nanostructures reflecting and scattering light and producing optical wave interference. All kinds of living organisms exhibit SC. However, understanding the molecular mechanisms and genes involved may be complicated due to the complexity of these organisms. Hence, bacteria that exhibit SC in colonies, such as Flavobacterium IR1, can be good models.Based on previous genomic mining and co-occurrence with SC in flavobacterial strains, this article focuses on the role of a specific gene, moeA, in SC of Flavobacterium IR1 strain colonies on an agar plate. moeA is involved in the synthesis of the molybdenum cofactor, which is necessary for the activity of key metabolic enzymes in diverse pathways.The authors clearly showed that the absence of moeA shifts SC properties in a way that depends on the nutritional conditions. They further bring evidence that this effect was related to several properties of the colony, all impacted by the moeA mutant: cell-cell organization, cell motility and colony spreading, and metabolism of complex carbohydrates. Hence, by linking SC to a single gene in appearance, this work points to cellular organization (as a result of cell-cell arrangement and motility) and metabolism of polysaccharides as key factors for SC in a gliding bacterium. This may prove useful for designing molecular strategies to control SC in bacterial-based biomaterials.Strengths:The topic is very interesting from a fundamental viewpoint and has great potential in the field of biomaterials.

Thank you for this.

The article is easy to read. It builds on previous studies with already established tools to characterize SC at the level of the flavobacterial colony. Experiments are well described and well executed. In addition, the SIBR-Cas method for chromosome engineering in Flavobacteria is the most recent and is a leap forward for future studies in this model, even beyond SC.

We appreciate these comments.

Weaknesses:The paper appears a bit too descriptive and could be better organized. Some of the results, in particular the proteomic comparison, are not well exploited (not explored experimentally). In my opinion, the problem originates from the difficulty in explaining the link between the absence of moeA and the alterations observed at the level of colony spreading and polysaccharide utilization, and the variation in proteomic content.

We have looked at the organisation of the manuscript carefully in this revision, as suggested. In terms of the proteomics, there are a large number of proteins affected by the *moe*A deletion and not all could be followed up. We chose spreading, structural colour formation and starch degradation to follow up phenotypically, as the most likely to be relevant. For example, (L615-617) we discuss the downregulation of GldL (which is known to be involved Flavobacterial gliding motility [Shrivastava et al., 2013]) in the *moe*A KO as a possible explanation for the reduced colony spreading of this mutant. Changes in polysaccharide (starch) utilization were seen on solid medium, as well as in the proteomic profile where we observed the upregulation of carbohydrate metabolism proteins linked to PUL (polysaccharide utilisation locus) operons (Terrapon et al., 2015), such as PAM95095-90 (Figure 8), and other carbohydrate metabolism-related proteins, including a pectate lyase (Table S7) which is involved in starch degradation (Aspeborg et al., 2012). And as noted in L555-566 and Figure 9, alterations in starch metabolism were investigated experimentally.

First, the effect of moeA deletion on molybdenum cofactor synthesis should be addressed.

MoeA is the last enzyme in the MoCo synthesis pathway, thus if only MoeA is absent the cell would accumulate MPT-AMP (molybdopterin-adenosine monophosphatase) (Iobbi-Nivol & Leimkühler, 2013), and the expressed molybdoenzymes would not be functional. In L582-585, we commented how the lack of molybdenum cofactor may affect the synthesis of molybdoenzymes. However, if you meant to analyse the presence of the small molecules, i.e. the cofactors involved in these pathways, that was an assay we were not able to perform. However, in L585-587, we addressed how the deletion of *moe*A affected the proteins encoded by the rest of genes in the operon which is relevant to the question.

Second, as I was reading the entire manuscript, I kept asking myself if moeA (and by extension molybdenum cofactor) was really involved in SC or it was an indirect effect. For example, what if the absence of moeA alters the cell envelope because the synthesis of its building blocks is perturbed, then subsequently perturbates all related processes, including gliding motility and protein secretion? It would help to know if the effects on colony spreading and polysaccharide metabolism can be uncoupled. I don't think the authors discussed that clearly.

The message of the paper is that the moeA gene, as predicted from a previous genomics analysis, is important in SC. This is based on the representation of the moeA gene in genomes of bacteria that display SC. This analysis does not predict the mechanism. When knocked out, a significant change in structural colour occurred, supporting this hypothesis. Whether this effect is direct or indirect is difficult to assess, as this referee rightly suggests. In order to follow up this central result, we performed proteomics (both intra- and extracellular). As we observed, the deletion of a single gene generated many changes in the proteomic profile, thus in the biological processes. Based on the known functions of molybdenum cofactor, we could only hypothesize that pterin metabolism is important for SC, not exactly how.

We have discussed the links between gliding/spreading and polysaccharide metabolism more clearly, with reference to the literature, as quite a bit is known here including possible links to SC.

“Polysaccharide metabolism in IR1 has been linked to changes in colony color and motility through the study of fucoidan metabolism (van de Kerkhof et al., 2022). Polysaccharide degradation and gliding motility are coupled to the same mechanism: the phylum-specific type IX secretion system, used for the secretion of enzymes and proteins involved in both functions (McKee et al., 2021).” [L622-626]

**Reviewer #2 (Public review):**
Summary:The authors constructed an in-frame deletion of moeA gene, which is involved in molybdopterin cofactor (MoCo) biosynthesis, and investigated its role in structural colors in Flavobacterium IR1. The deletion of moeA shifted colony color from green to blue, reduced colony spreading, and increased starch degradation, which was attributed to the upregulation of various proteins in polysaccharide utilization loci. This study lays the ground for developing new colorants by modifying genes involved in structural colors.Major strengths and weaknesses:The authors conducted well-designed experiments with appropriate controls and the results in the paper are presented in a logical manner, which supports their conclusions.

We appreciate these comments.

Using statistical tests to compare the differences between the wild type and moeA mutant, and adding a significance bar in Figure 4B, would strengthen their claims on differences in cell motility regarding differences in cell motility.

Thank you. Figure 4B contains the significance bars that represent the standard deviation of the mean value of the three replicates, but we have modified it to make them more clear.

Additionally, in the result section (Figure 6), the authors suggest that the shift in blue color is "caused by cells which are still highly ordered but narrower", which to my knowledge is not backed up by any experimental evidence.

Thanks. We mentioned that the mutant cells are narrower than the wild type based on the estimated periodicity resulting from the goniometry analysis (L427-430). We will now say “likely to be narrower based on the estimated periodicity from the optical analysis” rather than just “narrower”.

“This optical analysis aligns with visual observations, confirming the blue shift in Δ*moe*A, and suggests that this change in SC is caused by cells which are likely to be narrower based on the estimated periodicity from the optical analysis.” [L409-411]

Overall, this is a well-written paper in which the authors effectively address their research questions through proper experimentation. This work will help us understand the genetic basis of structural colors in Flavobacterium and open new avenues to study the roles of additional genes and proteins in structural colors.

Much appreciated.

**Recommendations for the authors:**

**Reviewing Editor Comments:**
As you will see, the reviewers were rather positive about the paper but suggested a number of points to improve it, including a discussion of the direct role of moeA as well as specific editorial comments.
**Reviewer #1 (Recommendations for the authors):**
More specific comments to the authors:(1) Line 300, Paragraph on bioinformatic analysis of molybdopterin operon :As written, it is not clear whether this operon is crucial for pterin cofactor synthesis or only some genes are involved. And what is the contribution of moeA?

Based on the bioinformatic analysis done in Zomer et al., 2024, we know the score of which genes of the molybdopterin cofactor synthesis operon may be more relevant to the display of SC, in addition to moeA. We chose moeA to KO as it had the highest score, being careful to delete the coding sequence and not any upstream promoter. The other genes in the predicted operon are *moa*E, *moa*C2, and *moa*A. Then in the proteomic analysis (L435-442), we analysed how the encoded proteins from this operon were upregulated (MoaA, MoaC2, and MobA), indicating also the unaltered proteins (MoeZ and MoaE) and the undetected proteins (MoaD and SumT). Nevertheless, the operon is crucial for pterin cofactor synthesis because it contains all the genes involved in the pathway, and *moe*A encoded the enzyme for the last reaction of the pathway, being the the molecule produced in the mutated pathway the adenylated molybdopterin (MPT-AMP) instead of molybdenum cofactor (MoCo).

(2) Paragraph line 342 on moeA mutant phenotyping :Is the reduction in colony spreading caused by a defect in single-cell gliding motility or is the cause more complex? This can be quantified.

We believe the cause is more complex. As mentioned above, for example, in (L615-617) we discuss the downregulation of GldL (which is known to be involved Flavobacterial gliding motility [Shrivastava et al., 2013]) in the *moe*A KO as a possible explanation for the reduced colony spreading of this mutant. This cannot be explained simply by spreading, but must (from the optical analysis) indicate changes in cell organisation/dimensions.

(3) During the description of the moeA mutant phenotype (associated with Figures 2 and 4) and throughout the article, the optical properties are « functions » of colony spreading and moeA-dependent metabolism. However it is not quite clear if these two effects are independent or if one may be a consequence of the other.

As noted above, colony spreading alone does not explain the blue-shift in SC observed. Given the function of MoeA (molybdate insertion into MPT-AMP [adenylated molybdopterin], MoMPT [molybdenum-molybdopterin] formation) for the synthesis of MoCo (molybdenum cofactor), the primary effect seems to be on metabolism but as we are dealing with an influential enzymatic cofactor a number of secondary effects are likely, and indeed the proteomics supports this. It is likely that the effect on spreading is secondary as seen with the downregulation of GldL (see above), but we cannot be sure.

(4) Paragraph starting line 381 and Figure 5 on gliding motility:Gliding motility has to be tested at the level of single cells, allowing a more thorough characterization of the spreading defects. In addition, since gliding is entangled with Type IX-dependent secretion in Flavobacteria, the authors should test if Type IXdependent was perturbed in the absence of moeA.

Based on the intracellular and extracellular proteomic analyses, the regulated T9SS proteins in the absence of *moe*A are the downregulation of GldL and SprT, and the upregulation of PorU. It shows the log2 FC (*moe*A/WT) of each these extracellular proteins:

**Author response table 1. sa3table1:** 

Protein	Intracellular	Extracellular
GIdL	0.34	-2.22
PorU	0.67	1.15
SprT	0.69	-

<-1: downregulated in moeA KO, -1<X<1: no significant regulation, >1: upregulated in *moe*A KO, -: not detected

(5) L401: In my opinion, the section "Quantification of the optical responses of IR1 WT and ΔmoeA colonies" should be moved up, before the characterization of motility.

We have done this, as suggested. The section was moved from L401-423 to L388-411.

(6) L475: Proteome comparison: « Of the total known proteins in IR1, 27.5% (1,504 proteins) extracellular proteins were identified » Are some of these proteins also found in the cell fraction? Wouldn't it be more accurate to write that « 1504 proteins were found in the extracellular fraction"?

We have done this, as suggested.

“Of the total known proteins in IR1, 27.5% (1,504 proteins) proteins were detected in the extracellular fraction, 60.4% (909) were statistically significant (p<0.01), with 20.5% (186) considered downregulated, and 20% (182) upregulated in Δ*moe*A (Figure 7B).” [L484-486]

How can the authors exclude contamination of the extracellular fraction? This could easily explain the number of proteins lacking secretion signals: "29.6% (55) were likely secreted through a non-classical way, lacking typical secretion sequence motifs in their N-terminus."

Based on the results from SecretomeP and SignalP, we excluded contamination, reducing the significant downregulated proteins from 186 (L476) to 69 (L486), and the upregulated ones from 182 (L477) to 111 (L500).

(7) L490: if the protein misannotated flagellin is highly downregulated, why not push the analysis a bit further and ask what true function may be perturbed? In addition, it should not be classified as a motility protein in Table S6 and considered as a motility protein in the article.

We reconsidered the information given by this and decided to remove it because after checking the homology of the polypeptide by Blast searching, we feel it is probably due to a missannotation.

As is, the whole proteomic section is not that useful. Too many functions are evoked and the reader is not directed toward any particular conclusion. The most convincing hits from the proteomic analysis should be confirmed using another method. Transcriptional regulation could be easily probed by RT-qPCR. Or, since genetics is possible, proteins could be tagged and levels compared by western blot maybe? Do knock-out of the encoding genes generate any phenotype on SC? This would bring weight to the proteomic analysis.

We have revised the proteomics section and removed functions that are not directly relevant to our conclusion.

We feel the most important observation suggested by proteomics was the possible link between moeA and starch metabolism, because the metabolism of complex polysaccharides is important in the *Flavobacteriia* and known to be linked to SC (van de Kerkhof et al., 2022). It was not possible to follow up every pathway suggested by the proteomics, but the study is appropriately performed with the correct statistics.

(8) Figure 9 : Does the absence of moeA affect the spreading of ASWS? Were colony sizes similar during the starch degradation assay? How can the authors rule out the idea that starch degradation is impacted by the difference in spreading rather than an independent function of moeA in starch metabolism? Slower spreading could lead to the accumulation of amylases, hence stronger activity. Why does starch degradation only accumulate at the center of the colony in the WT case?

The colonies of the WT and *moe*A had similar size during the starch degradation assay (2 days). However, after day 3, only WT colonies kept expanding on diameter.

Starch degradation is logically in the centre of the colony as it is where the greatest concentration of cells exists, secreting degradative enzymes, for the longest time. Presumably starch degradation at the colony edge is not yet seen as the action of extracellular enzymes is low and has not had time to degrade the starch to the point that there is no iodine staining.

“In contrast to other media where Δ*moe*A colony expansion was less than WT, the Δ*moe*A showed similar colony spreading and stronger starch degradation, supporting a role of *moe*A in complex polysaccharides metabolism.” [L562-565]

(9) Finally, I am not quite sure what the authors mean by « a role of moeA in complex polysaccharides metabolism ». Are they referring to enzymes secreted in the medium to degrade starch? or to the incorporation and use of starch degradation products?

We meant that the deletion of *moe*A showed an increase of extracellular starch degradation as seen in the iodine assay (Figure 9), as well as the upregulation of three different PUL operons (Figure 8).

**Reviewer #2 (Recommendations for the authors):**
The paper in general is well written with proper experimentation. However, here are a few recommendations for improving the writing and presentation, including minor corrections to the text and figures.

Thank you.

(1) It would be helpful for the readers if you could expand on "some metabolic pathways" in line 71. Please provide examples of metabolic pathways that are linked to SC.

We have done this.

“A recent bioinformatic study has shown the possible link of some metabolic pathways, such as carbohydrate, pterin, and acetolactate metabolism, to bacterial SC (Zomer et al., 2024).”[L70-72]

(2) "Line 79 : a bioinformatics analysis", please mention what kind of bioinformatics analysis was done and by whom to provide clarity for the readers: Either mention bio info analysis or give more details on what kind of bio info analysis and study done by whom"

We have clarified this, as suggested.

“A large-scale, genomic-based analysis of 117 bacteria strains (87 with SC and 30 without) identified genes potentially involved in SC by comparing gene presence/absence, providing a SC-score (Zomer et al., 2024). By this method, pterin pathway genes were strongly predicted to be involved in SC.” [L80-83]

(3) Please correct "Bacteria strains used in this study" to "bacterial" strains in Line 122.

We have done so.

(4) Please indicate in "Lines 394-396" that there were no vortex patterns observed in the moeA mutant.

We have done so.

“In contrast, Δ*moe*A exhibited limited motility, with a more tightly packed cell organization and a fine, slow-moving layer at the edge (Figure 6, blue arrows), and did not show a ‘vortex’ pattern. This suggests that *moe*A deletion significantly impairs cell motility and colony expansion.” [428-L431]

(5) In Figure 4 it looks like with a different carbon source (ASWB with agar and Fucoidan (ASWBF)) the moeA mutant and wild type exchanges its phenotype compared to ASWBKC. Could you explain why this happens in the discussion by highlighting the differences between fucose and Kappa-Carrageenan or confirm if there are any differences in the carbohydrate utilization between the wild type and moeA mutant using biolog assays?

We have explained the differences. Biolog would not be appropriate as we are looking for metabolic processes of bacteria on surfaces (agar) and this is not necessarily appropriate to biolog, which we understand uses liquid cultivation in microplates.

“On different polysaccharide media, the ΔmoeA strain showed varied SC and colony expansion patterns: green/blue SC and low colony expansion on agar, intense blue SC and low colony expansion on kappa-carrageenan, dull green SC and low colony expansion on fucoidan, and blue/green SC with higher colony expansion on starch. Interestingly, the color phenotype of the WT and Δ*moe*A exchanged their phenotype on kappa-carrageenan (a simple linear sulfated polysaccharide of D-galactopyranose) and fucoidan (a complex sulfated polysaccharide of fucose and other sugars as galactose, xylose, arabinose and rhamnose), showing the importance of the polysaccharide metabolism in SC. While reduced motility has been associated with dull or absent SC, and reduced polysaccharide metabolism (Kientz et al., 2012a; Johansen et al., 2018), Δ*moe*A showed reduced motility, but an intense blue SC, and high polysaccharide metabolism. Based on these results, we established a link among polysaccharide metabolism, MoCo biosynthesis, and SC, showing that intense SC is not strictly dependent on motility.” [L636-648]

(6) In the discussion "Line 632" it is unclear what loss is being limited, and it would help strengthen your discussion if you could add references for lines: 633-636. There are a lot of hypotheses in lines 637-642, it would help the readers if you could clearly mention that these are hypotheses and will need experimental evidence or provide appropriate evidence to support these claims.

We have done this.

“Ecologically, we hypothesize that dense, highly structured bacterial colonies, such as necessary for the SC phenotype, can enhance the uptake of metabolic degradation products from complex polysaccharides. These large macromolecules are often partially hydrolyzed extracellularly because they are too large to pass through bacterial cell membranes. For example, marine Vibrionaceae strains that produce lower levels of extracellular alginate lyases tend to aggregate more strongly, potentially facilitating localized degradation and uptake of polysaccharides (D’Souza et al., 2023). Additionally, certain marine bacteria employ a "selfish" mechanism to internalize large polysaccharide fragments into their periplasmic space, minimizing loss to the environment and enhancing substrate utilization (Reintjes et al., 2017). Bacteria secrete enzymes into the surrounding environment to break these polysaccharides down into more easily absorbable monosaccharides or oligosaccharides. This mechanism suggests that the colony structure could create a physical barrier that keeps these products concentrated and near the cells, allowing the colony to efficiently access and utilize these products, preventing the leakage into the surrounding environment. While SC may also yield other ecological benefits associated with growth in biofilms, the highly structured colonies that characterize SC may be more resistant against invasion by competitor species scavenging for degradation products, than an unstructured biofilm. This model is consistent with the observation that SC is associated with polysaccharide metabolism genes, and with the recent observation that SC is mainly localized on surface and interface environments such as airwater interfaces, tidal flats, and marine particles (Zomer et al., 2024).” [L650-670]

(7) It would help the readers if you could expand on how polysaccharide metabolism is linked to motility in Line 610.

As indicated previously, this is known and we will clarify.

“Polysaccharide metabolism in IR1 has been linked to changes in colony color and motility through the study of fucoidan metabolism (van de Kerkhof et al., 2022).” [L622-623]